# Revision of the Lichen Genus *Phaeophyscia* and Allied Atranorin Absent Taxa (Physciaceae) in South Korea

**DOI:** 10.3390/microorganisms7080242

**Published:** 2019-08-06

**Authors:** Dong Liu, Jae-Seoun Hur

**Affiliations:** Korean Lichen Research Institute (KoLRI), Sunchon National University, Suncheon 57922, Korea

**Keywords:** taxonomy, *Hyperphyscia*, *Physciella*, phylogeny

## Abstract

The genus *Phaeophyscia* Moberg, which belongs to the family Physciaceae, includes about 50 species, with 17 species reported in South Korea. This genus is characterized by a foliose thallus, *Physcia*/*Pachysporaria*-type ascospores, a paraplectenchymatous-type lower cortex, and lacking atranorin. In this study, about 650 specimens of *Phaeophyscia* aligned with the atranorin-absent groups collected from South Korea were re-examined. The taxonomy of these groups in South Korea requires revision based on the analyses of the morphology, chemistry, and molecular phylogeny. We infer that (1) each genus of the main foliose groups of Physciaceae forms a monophyletic clade, which also supports the separation of *Phaeophyscia* species with a prosoplectenchymatous lower cortex into the genus *Physciella*; (2) three atranorin-lacking genera were confirmed in South Korea: *Hyperphyscia*, *Phaeophyscia*, and *Physciella*, including a new combination named *Physciella poeltii* (Frey) D. Liu and J.S. Hur, and three new records from South Korea of *Phaeophyscia hunana*, *P. leana*, and *P. sonorae*; and (3) four species should be excluded from the lichen flora of South Korea: *Hyperphyscia adglutinata*, *Phaeophyscia endococcina*, *Phaeophyscia erythrocardia*, and *Phaeophyscia imbricata*.

## 1. Introduction

Lichens are one of the most successful symbiotes in nature, forming symbiotic associations with a fungus and algal partner, such as green algae and cyanobacteria. In general, lichens can be divided into fruticose, foliose, and crustose lichens based on the growth type. The foliose lichen genus *Phaeophyscia,* typed by *Phaeophyscia orbicularis* (Neck.) Moberg, belongs to the family Physciaceae and comprises ca. 50 species worldwide. This group usually occurs on bark, wood, bryophytes, rock, and soil in a wide range of habitats, and is characterized by a foliose thallus, white or orange-red medulla, paraplectenchymatous cortex, lecanorine apothecia with *Physcia*/*Pachysporaria*-type ascospores, ellipsoid conidia, and lacking atranorin [1,2,3,4,5,6,7,8,9].

*Phaeophyscia*, which was first described by Moberg, originated from the genus *Physcia* based on the ellipsoid conidia and lacking atranorin [1]. Later, Esslinger confirmed and extended Moberg’s delimitation on *Phaeophyscia*, for example, a paraplectenchymatous lower cortex, and *Physcia*/*Pachysporaria* type spore [2], and then placed the species with a prosoplectenchymatous lower cortex into a new genus *Physciella* with the following species: *Physciella denigrata*, *Physciella melanchra*, and *Physciella nepalensis* [10]. This definition, however, was not fully accepted by all lichenologists [4,5,8].

Taxonomical studies on *Phaeophyscia* have been frequently conducted around the world, from North America [2,5,11,12,13], East Africa [3], China [4,14,15], Japan [6,7,16,17,18,19], Europe [1,12,20,21,22,23,24] to Russia [4]. Little attention has been paid to the phylogenetics of this group, although some molecular studies of several species have been conducted by cooperation with the studies on *Physconia* and *Physcia*, *Anaptychia, Heterodermia*, and other relative groups [25,26,27,28,29].

South Korea is located in the southern half of the Korean Peninsula, and is dominated by mountains in the east (Gangwon province) and south and coastal plains and river tributary coastlines in the west. Since the first species of *Phaeophyscia endococcina* (= *P. endococcinodes*) was recorded in Korea [30,31], several species of the genus *Phaeophyscia* have been identified and reported since the 1990s. Park [32] reported 12 species with a simple description, and an increasing number of species were later found from the mainland and islands [33,34,35,36,37,38,39,40,41,42,43,44,45,46,47]. Hur listed 17 species of *Phaeophyscia* [46]; later, this number decreased to 16 species in Moon [44], and both publications recorded two species of *Physciella*. In 2015, a hairy species of *Phaeophyscia esslingeri* was discovered in Gangwon Province, a medium-latitude temperate zone mainly occupied by mountains [41]. Prior to this study, a total of three genera containing 25 species have been reported in South Korea, and these species have the following characteristics: A paraplectenchymatous upper cortex, lacking atranorin, and *Physcia-* or *Pachysporaria*-type ascospores.

In this study, we combined morphology, chemistry, and phylogenetic analysis with the aim of clarifying the relationship among the foliose genera in the Physciaceae family, and to investigate the constitution, distribution, and ecology of the atranorin-absent groups in Physciaceae more comprehensively.

## 2. Materials and Methods

### 2.1. Morphological and Chemical Studies

We examined approximately 650 specimens that were collected in South Korea from 2003 to 2017. All were deposited in the Korean Lichen Research Institute, Sunchon National University (KoLRI). The morphological characteristics and chemical spot tests were conducted under a dissecting microscope (Nikon SMZ 745T, Tokyo, Japan) and Olympus BX 50 microscope, and photographs were taken under a Carl Zeiss MicroImaging microscope with an Axio Cam ERc 5s imaging system (Carl Zeiss MicroImaging, GmbH 37081, Göttingen, Germany). All measurements based on the sections from the thalli and apothecia were recorded in water. The ascospore dimensions ranged from 20–80 spores from a single apothecium per specimen. The secondary metabolites were studied according to the spot tests in a solution (K = 10% aqueous KOH solution; C = saturated aqueous Ca(OCl)_2_; KC = 10% aqueous KOH solution followed by saturated aqueous Ca(OCl)_2_; P = 5% alcoholic p-phenylenediamine solution) and thin layer chromatography (TLC) in solvent C [48,49], and zeorine and atranorin were selected as controls.

### 2.2. DNA Isolation, PCR, and DNA Sequencing and Alignment

The total genomic DNA was extracted from the specimens using the NucleoSpin Plant II Kit (Clontech Laboratories, Mountain View, CA, USA) according to the manufacturer’s instructions. The internal transcribed spacer (ITS) region was generated using the primers pairs ITS1F [50] and ITS4 [51]. We followed the protocols for amplification and sequencing by PCR [52]. Sequencing was conducted by the genomic research companies GenoTech (Daejeon, Korea) and Macrogen (Daejeon, Korea). Newly generated ITS sequences were complemented with the published sequences of *Physciaceae* from GenBank (Appendix A). All raw sequences were assembled and edited using SeqMan (DNAstar packages, Madison, WI, USA) and BioEdit 7.09 (Carlsbad, CA, USA) [53], then aligned automatically with Mafft version 7.273 (Osaka, Japan) [54].

### 2.3. Phylogenetic Analysis

The maximum likelihood (ML) optimality criterion and Bayesian inferences (BI) were used to construct the phylogenetic trees. ML inferences were performed using RAxML v7.2.6 (Heidelberg, Germany) [55], using the GTR model. The bootstrap frequencies were estimated from the consensus tree built with 2000 trees obtained from nonparametric bootstrapping pseudoreplicates. Clades with an ML bootstrap value ≥70% were strongly supported. BI analyses were performed with MrBayes v3.1.2 (San Francisco, CA, USA) [56] using four chains and were run for 1 million generations. SYM + I + G were selected as the best-fitted substitution models based on the Akaike information criterion (AIC) using jModelTest 3.7 (San Francisco, CA, USA) [57]. The trees were sampled every 1000 generations. The phylogenetic trees were summarized using the sump and sumt commands with 20% burn-ins discarded. The Bayesian posterior probabilities (PP) were estimated from the frequencies of branches among all trees; clades with PP ≥ 0.95 were considered as being significantly supported. A phylogenetic tree was rooted with the *Heterodermia* species.

## 3. Results

### 3.1. Genera Relationship

The ML consensus tree combined bootstrap and PP values on the nodes is presented in Figure 1. Each genus was represented by several species, which are not shown in the figure. The phylogenetic tree was rooted by the genus *Heterodermia*, and each clade inferring to one genus formed a single clade with high support. The genus *Phaeophyscia* is sister to *Physciella* and other atranorin-absent groups (e.g., *Hyperphyscia*, *Anaptychia*, and *Physconia*). All of the atranorin-absent groups with K− formed a big clade and are separate from the atranorin groups (*Heterodermia* and *Physcia*) with K+. The genus *Physconia* and *Anaptychia* are closely related to the genus *Hyperphyscia*. All these genera form a monophyletic group.

### 3.2. Species Relationship of the Atranorin Absent Groups

Figure 2 shows the phylogenetic tree with the species information. Seven genera with 42 species were included in the topology, which contained *Phaeophyscia* (17 sp.), *Physciella* (3 sp.), *Hyperphyscia* (3 sp.), *Anaptychia* (4 sp.), *Physconia* (5 sp.), *Physcia* (6 sp.), and *Heterodermia* (4 sp.). The Korean specimens were placed into three genera. *Hyperphyscia crocata* was strongly supported as belonging to *Hyperphyscia*, whereas *Physciella melanchra* and *Physciella poeltii* (= *Phaeophyscia poeltii*) were placed into *Physciella*. The specimens with a paraplectenchymatous lower cortex belonged to *Phaeophyscia*, but only a few internodes were highly supported.

### 3.3. Taxonomy

Three genera (including 18 species), *Hyperphyscia*, *Phaeophyscia*, *Physciella*, were confirmed; these contained 17 species from South Korea and one species from Europe. Among them, three species, *Phaeophyscia hunanna*, *Phaeophyscia leana*, and *Phaeophyscia sonorae*, are new to South Korea; four species, *Hyperphyscia adglutinata*, *Phaeophyscia endococcina*, *P. erythrocardia*, and *Phaeophyscia imbricata*, were excluded from the lichen flora of South Korea. In addition, one combination, *Physciella poeltii*, was proposed based on the molecular and morphological characteristics. All the specimens examined are listed in Appendix A.

        **Key to the foliose genera in Physciaceae.**

1a. Spore *Physconia* type ………………………………………… 2

1b. Spore *Physcia* or *Pachysporaria* type …………………… 3

2a. Upper cortex prosoplectenchymatous ……………… *Anaptychia*

2b. Upper cortex paraplectenchymatous ……………… *Physconia*

3a. Thallus K+ yellow, containing atranorin ……………… 4

3b. Thallus K−, lacking atranorin ………………………… 5

4a. Upper cortex prosoplectenchymatous ……………… *Heterodermia*

4b. Upper cortex paraplectenchymatous ……………… *Physcia*

5a. Conidia filiform, usually more than 10 µm long, lower cortex not indistinct or prosoplectenchymatous, erhizinate or with short sparse inconspicuous rhizines …………… *Hyperphyscia*

5b. Conidia ellipsoid, less than 6 µm long, rhizinate or with distinct lower cortex ………… 6

6a. Lower cortex prosoplectenchymatous ……………… *Physciella*

6b. Lower cortex paraplectenchymatous ……………… *Phaeophyscia*

#### 3.3.1. Species New to South Korea

##### *Phaeophyscia hunana* G.R. Hu & J.B. Chen

Thallus foliose, brown to dark brown, lobate, closely adnate substrate, up to 2 cm in diameter; lobes contiguous or imbricate, variable in width, 0.3–0.8 (–1) mm wide; thallus 90–120 µm thick; upper surface plane or more or less convex, epruinose, lacking isidia, soredia, and pruina, lobules and cortical hair occasionally present; upper cortex paraplectenchymatous, 20–30 µm thick; algal layer continuous, 25–30 µm thick; medulla orange-red, loose, 35–75 µm thick; lower surface black, rhizinate; rhizines dense, black; lower cortex paraplectenchymatous, brown, 15–30 µm thick; apothecia not seen (Figure 3B).

Chemistry: Thallus K−; medulla K+ violet; skyrin and unidentified substance (Rf = 6, in C system) present.

Ecology: The species is found growing on bark.

Distribution: *Phaeophyscia hunana* was recorded in China [14,15], and here reported for the first time in South Korea.

Comments: This species is characterized by a small thallus with narrow lobes, lobules, orange medulla, and black lower surface, lacking atranorin, and containing skyrin and an unidentified substance.

This species is similar to *P. fumosa* Moberg, but differs in having lobules and in containing an unidentified substance; *P. hunana* resembles *P. laciniata*, a species that also has lobules, but the latter species has a much larger size and saxicolous. *P. laciniata* lacks cortical hairs on the thallus or lobe tips.

Representative specimens examined: Jeju Province, Jeju-si, Mt. Halla, Seongpanak Trail, 33°22’48" N, 126°35’27" E, a.s.l. 1025 m, on bark, July 6, 2012, S.Y. Kondratyuk et al. 121894.

##### *Phaeophyscia leana* (Tuck.) Essl

Thallus foliose, white to pale grey, lobate, up to 3 cm in diameter (diameter); lobes elongate, strap-shaped, irregularly branched, contiguous or imbricate, 0.3–1.2 mm wide; upper surface flat to more or less convex, slightly maculate, lacking soredia, isidia, and lobules; medulla white; lower surface pale to white, rhizinate; rhizines dense, concolor with lower surface, up to 3 mm long; thallus 150–180 µm thick; upper cortex paraplectenchymatous, 23–30 µm thick; algal layer continuous, approximately 20–30 µm thick, algae cell 6–10 µm in diameter; medulla loose, 50–100 µm thick; lower cortex paraplectenchymatous, 15–25 µm thick; apothecia common, cup-shaped, stipitate, up to 1.2 mm in diameter; margin entire or weakly crenate, without cortical hair; disc reddish brown, epruinose; amphithecium covered with retrorse white to black rhizines near the base, sparse; hymenium 80–100 µm thick; hypothecium pale brown, 60–80 µm thick; asci clavate, 8-spored, 60–75 × 10–15 µm; spores brown, *Physcia* type, (16.1–)16.2–*18.9*–20.8(–23.5) × (7.5–)7.7–*8.9*–12(–12.2) µm, Q = 1.5–2.5, Q(m) = 2.1, n = 71; pycnidia common, brown to black spotted; conidia 2–3 × 1–1.5 µm (Figure 3A,C–F).

Chemistry: Thallus K−; medulla K−, C−, KC−, P−; no lichen substance detected.

Ecology: This species was only found growing on bark.

Distribution: *Phaeophyscia leana* was reported in North America [2] and newly reported in South Korea.

Comments: *Phaeophyscia leana* is characterized by a slight maculate gray thallus, lacking asexual propagules, apothecia with sparse rhizines near the base, white medulla, pale to white lower surface with a paraplectenchymatous cortex, and an absence of lichen substances.

*Phaeophyscia leana* is most similar to *P. hirtella* and *P. hirtuosa*, but can be distinguished from the latter species by a white lower surface. This species resembles *P. trichophora* (Hue) Essl., but can be distinguished by the Physcia type spores, stipitate apothecia with sparse rhizines near the base, whereas *P. trichophora* frequently has sessile and strongly crenate apothecia with numerous rhizines near the base as well as *Pachysporaria*-type spores [2]. *Phaeophyscia leana* differs from *P. sonorae* by having rhizines near the apothecia base, and from *P. constipate* by having a foliose thallus and usually occurs on bark, whereas the latter species are usually fruticose and mixes with moss over soil.

Specimen examined: Gangwon, Wonju, Socho, Mt. Chiak, 37°16′60″ N, 128°01′05″ E, 465–480 m, on bark, August 13, 2004, J.S. Hur 040554.

##### *Phaeophyscia sonorae* Essl.

Thallus foliose, grey to dull, loosely attached substrate, 5–8 cm in diameter; lobes irregularly or dichotomously branched, imbricate, more or less concave, 0.5–1.8 mm wide; upper surface without soredia, isidia, and lobules with pruinose-like pachy; medulla white; lower surface white to pale tan, rhizinate; rhizines simple, concolor with lower surface or darken, up to 3 mm long; thallus 220–250 µm thick; upper cortex paraplectenchymatous, 15–25 µm thick; algal layer 50–70 µm thick, algae cells green, 5–12 µm in diameter; medulla loose, 90–120 µm thick; lower cortex paraplectenchymatous 20–25 µm thick; apothecia not seen; pycnidia common, immersed into thallus, flesh to black; conidia ellipsoid, 2.5–3 × 1–1.5 µm.

Chemistry: Thallus K−; medulla K−; lichen substance absent.

Ecology: The species is found growing on bark.

Distribution: *Phaeophyscia sonorae* is distributed in North America and Sonora [5], and here for the first time reported in South Korea.

Comments: This species is characterized by a grey thallus without soredia and isidia, medulla white, lower surface pale or white with concolor rhizines, and a paraplectenchymatous lower cortex. 

*Phaeophyscia sonorae* is similar to *P. leage*, but lacks patchiness over the upper surface. This species resembles *Physciella nepalensis* and *P. denigrata* but differs in having a paraplectenchymatous lower cortex. *Phaeophyscia sonorae* can be distinguished from *P. nashii* by lacking soredia.

Representative specimens examined: Gyeongsangnam Province, Hamyang County, Seosang Town, Mt. Baekun, 35°36′19″ N, 127°39′36″ E, 917 m, on *Quercus*, June 24, 2010, X.Y. Wang et al. 100376.

#### 3.3.2. New Combination

*Physciella poeltii* (Frey) D. Liu and J.S. Hur, comb. nov.

*Physcia poeltii* Frey, Ber. Schweiz. Bot. Ges. 73: 490 (1963)

Comments: *Physciella poeltii* is a rare species known from Switzerland, Italy, Bulgaria, and Yugoslavia, occurring on the trunks of deciduous trees in open areas, and is characterized by a grey brown thallus, whitish lower surface, paraplectenchymatous upper cortex, and prosoplectenchymatous type lower cortex, *Physcia*/*Pachysporaria*-type ascospores, ellipsoid conidia, and lacking substances [20].

#### 3.3.3. Recorded Species

##### *Hyperphyscia crocata* Kashiw.

Thallus foliose, 2–4 cm diameter, lacking a hypothallus, closely adnate to the substratum; lobes 0.8–1.5 mm wide, often crowded or imbricated, broadened at the tip and blackened at the lobe tip margin; upper cortex grey-green to olive or brown, sorediate, soralia laminal, maculiform, occasionally orange, marginal or submarginal, frequently dense in the central part thallus; medulla golden to orange or white with an orange band in the lower medulla; lower cortex indistinct or not apparent; lower surface black, pale to brown at the lobe tips; apothecia and pycnidia not seen.

Chemistry: Upper cortex K−, KC−; atranorin absent, skyrin present.

Ecology: This species was found growing on trees, and together with *Mikhtomia geumohdoensis* and *Dirinaria applanata*.

Distribution: The species has only been reported in Japan and South Korea.

Comments: *Hyperphyscia crocata* is characterized by an appressed thallus without black hypothallus, contiguous lobes with orange medulla, laminal soralia, pale to brown lower surface lobe tip, and black lower surface in the center parts of the thallus. *Hyperphyscia crocata* is most similar to *H. pandani*, which has a wide distribution in North America, East Africa, and Australia [58,59]; they share similar orange medulla and a thallus with soralia, whereas the type specimen of *H. pandani* has a black hypothallus and narrower lobes (0.5–1 mm). *Hyperphyscia crocata* has been reported primarily from Jeju Island [33], and is also distributed on the mainland [45].

Representative specimens examined: Jeollanam Province, Yeosu City, Nam Town, Dumo Road, Jickpo Coast, Geumoh-do, 34°43′10″ N, 127°37′21″ E, 6 m, on bark, April 26, 2012, S.Y. Kondratyuk 120413.

##### *Phaeophyscia adiastola* (Essl.) Essl.

Thallus greenish gray to dark gray or brownish, up to 5 cm in diameter, orbicular to irregular; lobes irregularly branched, more or less linear 0.5–2 mm broad, flat or often concave, frequently upturned toward tips; upper surface sorediate, soralia irregular to weakly capitate sometimes, terminal or marginal; soredia coarsely granular to isidioid; medulla white; lower surface black, often paler on the lobe tips, covered with numerous long black rhizines; upper, and lower cortex paraplectenchymatous; apothecia occasionally not seen.

Chemistry: Thallus K−; medulla K−, C−, KC−, P−; no substance detected.

Ecology: This species is found frequently growing on bark or sometimes moss over bark.

Distribution: *Phaeophyscia adiastola* is distributed in North America, China, and South Korea.

Comments: *Phaeophyscia adiastola* is characterized by its more or less linear lobes and its granular to isidioid soredia and marginal or terminal soralia.

Representative specimens examined: Gangwon Province, Hongcheon County, Nam- County, Mt. Eungbok, 37°51′47″ N, 128°33′32″ E, 866 m, on bark, May 23, 2009, Y. Joshi et al. 090741. Gyeongsangnam-do. Geochang-gun, Wicheon-myeon, Mt. Geumwon, 35°43′39″ N, 127°45′50″ E, 1351 m, on bark, June 25, 2010, X.Y. Wang et al. 100555. Jeollabuk Province. Gochang County, Asan Town, Mt. Seonun, 36°57′27″ N, 128°26′36″ E, May 11, 2003, J.S. Hur 030246. Jeollanam Province. Gurye County, Toji Town, Mt. Jiri, 35°19′14″ N, 127°39′46″ E, 1491 m, on bark, September 29, 2006, J.S. Hur 060923.

##### *Phaeophyscia endococcinodes* (Poelt) Essl.

Thallus foliose, white to gray or greenish brown, lobate, 1.5–3.5 cm in diameter; lobes contiguous or imbricate, variable in width, 0.2–0.8 mm wide; thallus 90–150 µm thick; upper surface plane or more or less convex, epruinose, lacking isidia, soredia, and pruina; upper cortex paraplectenchymatous, 20–35 µm thick; algal layer continuous, 25–40 µm thick; medulla orange-red, loose, 40–85 µm thick; lower surface black, rhizinate; rhizines dense, black, 0.4–2.5 mm long; lower cortex paraplectenchymatous, brown, 20–40 µm thick; apothecia rare to numerous, up to 1 mm in diameter; margin entire, or with stellate lobules; disc reddish brown, epruinose; amphithecium retrorsely hairy at the base; cortex of apothecia composed of 3–4 layers of larger cells (7–11 µm in diameter) without outer an amorphous layer; hymenium 80–110 µm high; hypothecium hyaline or pale brown, 25–40 µm thick; asci cylindrical, 55–80 × 9–12 µm; spores brown, *Pachysporaria* type, 17–25 × 8–10 µm; pycnidia immersed in thallus, brown to black; conidia 2.5–3.5 × 1–1.5 µm.

Chemistry: Thallus K−; medulla K+ violet; skyrin present, zeorine occasionally present.

Ecology: The species is only found growing on rocks.

Distribution: *Phaeophyscia endococcinodes* is distributed widely in Japan, China, North America, Russia, East Asia, and South Korea [2,3,4,18,31], distribution in South Korea.

Comments: This species is characterized by the narrow lobes (<1.5 mm), lacking asexual propagules, a black lower surface with a paraplectenchymatous lower cortex, the spores of the *Pachysporaria* type, and containing skyrin and occasionally zeorine.

*Phaeophyscia endococcinodes* is related to *P. endococcina*, which is common in North America, the Himalayas, Europe, and East Africa [2,3,18], but *P. endococcina* has *Physcia*-type spores, *P. endococcinodes* has spores of the *Pachysporaria* type, and the zeorine is consistently present in *P. endococcina*, but trace or absent in *P. endococcinodes*. This species may be also confused with *P. erythrocardia*, from which it can be distinguished by the cortex of amphithecium composed of bigger cells (more than 7 µm in diameter) [2,18], and the latter species has the spore of the *Physcia* type. *Phaeophyscia endococcinodes* can be distinguished from *P. ciliata* by the orange-red medulla containing skyrin.

The present species has been reported to have a corticolous and broader lobe from South Korea [60], but should be *P. pyrrhophora* based on an examination of these specimens. In addition, the specimen (Hur 030833) reported as *P. erythrocardia* [60] should be *P. endococcina* because no zeorine was detected and it had stellata lobules around the amphithecium [18] as well as *Pachysporaria*-type spores.

Representative specimens examined: Chungcheongnam Province. Seosan City, Buseok Town, Buseok Temple, 36°57′33″ N, 128°29′18″ E, 1364 m, on rock, October 4, 2003, J.S. Hur 030833. Jeollanam Province, Sinan County, Bigeum Town, Mt. Sunwang, Bigeum island, 34°44′14″ N, 125°55′48″ E, 92 m, on rock, June 5, 2013. S.O. Oh et al. 130163.

##### *Phaeophyscia exornatula* (Zahlbr.) Kashiw.

Thallus foliose, greenish grey, more or less loosely attached to the substrata, 3–12 cm in diameter; lobes irregularly branched, imbricate, upturned near the tips, concave, 1.5–3 mm wide, lacking cortical hairs on the upper surface, lobulate; lobules fragile, dorsiventral, developed from submarginal pustules, or break into soralia or isidioid soralia or lobulated isidia, usually bearing black rhizines on the lower surface or on the margin, never producing cortical hairs on the upper surface of the lobules, 200–600 µm wide; medulla white; lower surface black, rhizinate; rhizines dense, black, simple, up to 3 mm long; thallus 80–220 µm thick; upper cortex paraplectenchymatous, 20–35 µm thick; algal layer 30–35 µm thick, algae cells 5–14 µm in diameter; medulla 50–100 µm thick; lower cortex dark brown, paraplectenchymatous, 15–50 µm thick; apothecia up to 3 mm in diameter; margin entire or crenate, often with lobules, lacking cortical hairs; disc reddish brown, flat to concave; amphithecium retrorsely haired near the base; hymenium 75–90 µm high; hypothecium very pale brown, 50–60 µm thick; asci clavate, eight-spored, 70–80 × 8–14 µm; spores brown, *Physcia* type, (16.3)16.7–*19.2*–21(22.8) × (8)9.1–*10.1*–11.8(11.9) µm, Q = 1.7–2.0, Q(m) = 1.9, n = 25; pycnidia common, immersed into thallus, black; conidia ellipsoid, 2.5–3 × 1–1.5 µm.

Chemistry: Thallus K−; medulla K−, C−, KC−, P−; no substances detected.

Ecology: This species is found frequently growing on bark, moss over bark, and rock.

Distribution: *Phaeophyscia exornatula* is distributed in Japan and South Korea [17,60].

Comments: This species is characterized by concave, imbricate lobes, the presence of dorsiventral lobules and usually break into isidoid soralia or lobulated isidia, the absence of cortical hairs on the upper surface of the lobes, and on the apothecial margin, white medulla, *Physcia*-type spores, and absence of lichen substance.

*Phaeophyscia exornatula* resembles *P. kairamoi* (Vain.) Moberg, a species reported in North America and Europe, both have similar lobules or lobulated isidia, but the first differs from the latter by the absence of cortical hairs, which are found on the upper surface of lobes or on lobules in *P. kairamoi*. The species can be distinguished from *P. sciastra* (Ach.) Moberg by dorsiventral lobules and broader lobes, and differs from *P. squarrosa* by lacking zeorin.

Representative specimens examined: Chungcheongbuk-do. Danyang-gun, Gagok-myeon, Mt. Sobaek: 36°57′33″ N, 128°29′18″ E, 1375 m, on bark, October 2, 2003, J.S Hur 030763. Gyeongsangbuk Province. Bonghwa-gun, Mulya-myeon, Mt. Seondal: 37°01′03″ N, 128°43′19″ E, 485 m, on saxicolous mosses, April 24, 2007, J.S Hur 070216. Gyeongsangnam Province. Hamyang County, Macheon Town: Changwon road, Mt. Baekun, 35°38′18″ N, 127°37′27″ E, 1087 m, on trunk, July 4, 2015, S.O. Oh et al. 150747. Incheon Metropolitan City. Ganghwa County, Samsan Town, Seokmo Island, 37°41.381’ N, 126°19.320′ E, 160 m, on saxicolous mosses, September 29, 2010, X.Y. Wang et al. 101045. Jeollanam Province. Gurye County, Masan Town, Mt. Jiri, Hwaeom Valley, 35°16.905′ N, 127°31.003′ E, 816 m, on bark, October 12, 2009, Y. Joshi et al. 091045, 091055.

##### *Phaeophyscia hirtella* Essl.

Thallus greenish gray to darker gray or slight brownish, 1–5 cm in diameter, orbicular or irregular, more or less close adnate to the substrate; lobes dichotomously or irregularly branched, irregularly rounded to more or less linear, usually flat, 0.3–1.5 mm broad, with sparse to numerous, pale or almost hyaline cortical hairs on lobe ends, margin or surface; thallus 150 µm thick; upper surface often distinctly white maculate, without soredia, isidia, or lobules; upper cortex paraplectenchymatous, 25–35 µm thick; algal layer continuous, approximately 30 µm thick, algae cells 7–11 µm in diameter; medulla white, loose, 70–90 µm thick; lower cortex paraplectenchymatous, blackish brown, 20–30 µm thick; lower surface black, sometimes paler on the lobe ends, with rather numerous black rhizines; apothecia numerous, up to 3 mm in diameter, often with darkened rhizines around the base, and bearing pale or darken cortical hairs around the margin, 50–300 µm long; hymenium 70–80 µm high; hypothecium hyaline or very pale straw-yellow, 30–45 µm thick; asci clavate, eight-spored, 70–80 × 13.4–15 µm; spores brown, *Physcia* type, (16.1)16.5–*19.1*–21(23.7) × (6.43)7.8–*9.1*–10.63(10.8) µm, Q = 1.5–2.7, Q(m) = 2.1, *n* = 51; pycnidia common, immersed into thallus, black; conidia ellipsoid, 2.5–3 × 1–1.5 µm.

Chemistry: Thallus K−; medulla K−, C−, KC−, P−; lichen substance absent.

Ecology: This species is found frequently growing on the bark of *Quercus* spp. or sometimes on moss over bark.

Distribution: *Phaeophyscia hirtella* is distributed in North America and South Korea [2,60].

Comments: *Phaeophyscia hirtella* is characterized by a whitish grey thallus without asexual propagules, slight maculate lobes with cortical hair near the margin and ends, black lower surface with black rhizines, white medulla, apothecia with numerous cortical hairs on the margin and rhizines near the base, *Physcia*-type spores, and an absence of lichen substance.

*Phaeophyscia hirtella* can be confused easily with *P. hirtuosa* (Krempelh,) Essl., a widely distributed species in Asia and North America, but differs from the latter species in having numerous cortical hairs near the lobe margin or ends, and having broader lobes. This species is similar to *P. ciliata*, but *Phaeophyscia ciliata* differs from *P. hirtella* in lacking cortical hairs. In addition, *P. hirtella* is corticolous.

Representative specimens examined: Chungcheongbuk Province. Danyang County, Gagok Town, Mt. Sobaek, on bark, October 2, 2003, J.S. Hur 030738. Gangwon-do. Gyeonggi Province. Gapyeong County, Buk Town, Mt. Myeongji, 37°55′58″ N, 127°28′53″ E, 223 m, on *Salix*, September 20, 2008, J.S. Hur 080666. Gyeongsangbuk Province. Bonghwa County, Mulya Town, Mt. Seondal, 37°00′58″ N, 128°43′21″ E, 278 m, on bark, April 24, 2007, J.S. Hur 070211. Incheon Metropolitan City. Ganghwa County, Hwado Town, 37°42′41″ N, 126°23′27″ E, 52 m, on *Zelkova*, September 30, 2010, 101132. Jeju-Do. Jeju City, Arail, Sancheon, 33°26′50″ N, 126°33′15” E, 371 m, on bark, December 17, 2015, J.S. Park 152879.

##### *Phaeophyscia hirtuosa* (Kremp.) Essl.

Thallus foliose, greenish gray, lobate, up to 6 cm in diameter; lobes irregularly branched, contiguous or imbricate, 1–3.5 mm wide; upper surface flat to more or less concave, lacking soredia, isidia and lobules; medulla white; lower surface black, rhizinate; rhizines dense, black, often whitish at the tips, projecting beyond the margins of the lobes, up to 5 mm long; thallus 120–250 µm thick; upper cortex paraplectenchymatous, 20–25 µm thick; algal layer continuous, approximately 25–30 µm thick, algae cell 7–11 µm in diameter; medulla loose, 100–200 µm thick; lower cortex paraplectenchymatous, dark brown, 20–30 µm thick; apothecia common, cup-shaped, up to 4.5 mm in diameter; margin entire or crenate, with cortical hair; disc reddish brown, epruinose; amphithecium covered with retrorse hairs; hymenium 78–90 µm thick; hypothecium pale brown, 30–50 µm thick; asci clavate, eight-spored, 60–80 × 10–20 µm; spores brown, *Physcia* type, (17)18.1–*19.6*–22.8(23.2) × (7.5)7.7–*9.4*–10.6(10.8) µm, Q = 1.7–2.5, Q(m) = 2.1, *n* = 75; pycnidia common, flesh to black spotted; conidia 2–3 × 1–1.5 µm.

Chemistry: Thallus K−; medulla K−, CV−, KC−, P−; no lichen substance detected.

Ecology: This species is only found growing on bark.

Distribution: *Phaeophyscia hirtuosa* is widely distributed in Japan, China, North America, Russia, and South Korea [2,4,19,33,60].

Comments: This species is characterized by the thallus lacking asexual propagules, apothecia margin covered with cortical hairs and rhizines near the base, white medulla, black lower surface, and an absence of lichen substances.

*Phaeophyscia hirtuosa* is most similar to *P. hirtella*, but can be distinguished from the latter species by the following characteristics: The absence of cortical hairs on the lobe or lobe margin or tips, whereas they are usually present in the latter species; in addition, the lobe of *P. hirtuosa* is much broader. This species resembles *P. spinellosa*, but differs in having larger spores and corticolous.

Representative specimens examined: Jeju Province, Jeju City, 33°27’15"N, 126°33’41"E, 370 m, on bark, August 29, 2004, J.S. Hur 040859.

##### *Phaeophyscia hispidula* (Ach.) Essl.

Thallus grey, orbicular to irregular, up to 7 cm in diameter; lobes irregularly branched, contiguous or imbricate, flat to concave, 1–3 mm wide; upper surface sorediate, soralia laminal, starting as pustules and sometimes reaching the margins, then becoming capitate; medulla white; upper surface black, covered with long black rhizines; upper and lower cortex paraplectenchymatous, cells isodiametric with dark brown walls in the lower cortex; apothecia not seen.

Chemistry: Thallus K−; medulla K−, C−, KC−, P−; no lichen substance detected.

Ecology: This species is found frequently growing on bark, and occasionally on rocks.

Distribution: *Phaeophyscia hispidula* is widely distributed in Japan, China, North America, and South Korea.

Comments: *Phaeophyscia hispidula* is characterized by broad lobes with capitate marginal or laminal soralia, white medulla, and a black lower surface.

Representative specimens examined: Gangwon Province. Jeongseong County, Gangneung City, tourist pass toward peak Seokbyeongsan, 37°34′37″ N, 128°51′47″ E, 840 m, on bark, July 10, 2015, J.J. Woo et al. 151051. Gyeongsangnam Province. Hamyang County, Seosang Town, Mt. Baekun, 35°36′20″ N, 127°39′38″ E, 917 m, on *Quercus*, June 24, 2010, X.Y. Wang et al. 100400. Jeollabuk Province. Muju County, Seolcheon Town, Mt. Sambong, 35°52′08″ N, 127°49′42″ E, 970 m, on *Quercus*, June 18, 2015, J.J. Woo et al. 150046. Jeollanam Province. Gwangyang City, Okryong Town, Mt. Baekun, 35°04′39″ N, 127°39′38″ E, 675 m, on *Quercus*, September 25, 2004, J.S. Hur 041254.

##### *Phaeophyscia limbata* (Poelt) Kashiw.

Thallus foliose, greenish grey, 2–10 cm in diameter; lobes irregularly or dichotomously branched, discrete or imbricate, sorediate, 2–4 mm wide; upper surface concave, without cortical hairs; soralia marginal or sub-marginal, often spreading over the upper surface, linear; soredia granular, elongate to isidioid; medulla white; lower surface black, rhizinate; rhizines black, simple; thallus 150–220 µm thick; upper cortex paraplectenchymatous 15–25 µm thick; algal layer continuous, 25–40 µm thick; medulla 50–130 µm thick; lower cortex paraplectenchymatous, blackish brown, 25–50 µm thick; apothecia rare, up to 3 mm in diameter; margin entire, lacking cortical hairs, rarely sorediate with granular soralia; disc reddish brown; amphithecium with retrorsely hair near the base; hymenium 80–95 µm high; hypothecium very pale brown, 40–70 µm thick; asci clavate, 8-spored, 68–88 × 13–26 µm; spores brown, *Physcia* type, (17.8)19.2–22.6–25.1(25.8) × (7.5)8.5–11.1–13.6(15.5) µm, Q = 1.6–2.4, Q(m) = 2.0, *n* = 27; pycnidia common, immersed into the thallus, brown to black; conidia ellipsoid, 2.5–3 × 1–1.5 µm long.

Chemistry: Thallus K−; medulla K−; lichen substance absent.

Ecology: The species is found growing on bark, moss, or rock.

Distribution: *Phaeophyscia primaria* is distributed widely in China, Nepal, Japan, and South Korea [4,6,32].

Comments: This species is characterized by the distinctly concave, branched, and broad lobes (2–4 mm wide) without cortical hairs, marginal or sub-marginal granular to isidioid soredia, black lower surface with simple black rhizines, white medulla, apothecial margin without cortical hairs, *Physcia*-type spores, and an absence of lichen substance. The soredia of this species are variable, ranging from granular, elongate to isidiod, but usually numerous growing on the lobe margin, and if spread to the upper surface, it usually presents frequently on the young lobes and is much smaller than elongate or isidiod ones.

*Phaeophyscia limbata* differs from *P. hispidula* by the marginal elongate soralia instead of the capitate soralia. The specimens with coarse isidioid soredia of this species are confused easily with *P. exornatula,* from which it can be separated by the lack of dorsiventral lobules. The species also resembles *P. adiastola* (Essl.) Essl., a species described from North America, but can be separated from the latter species by a continuous and robust thallus and broader lobes, whereas those of *P. adiastola* are discrete and relatively slender.

Representative specimens examined: Chungcheongbuk Province. Danyang County, Gagok-myeon, Mt. Sobaek: 36°57′27″ N, 128°26′36″ E, 594 m, October 1, 2003, J.S. Hur 030708, 030720. Chungcheongnam-do. Gongju-si, Banpo-myeon, Mt. Gyeryong, 36°21′47″ N, 127°13′30″ E, 440 m, on *Quercus*, October 23, 2004, J.S Hur 041617. Gangwon Province. Chuncheon City, Buksan Town, Jogyo Road, Mt. Maebong, 37°54′40″ N, 127°58′58″ E, 610 m, on *Quercus*, May 25, 2010, X.Y. Wang et al. 100566, 100569, 100594. Gyeongsangbuk-do. Bonghwa County, Mulya Town, Mt. Seondal, 37°01′06″ N, 128°43′03″ E, 520 m, on bark, April 24, 2007, J.S Hur 070220. Gyeongsangnam Province. Geochang County, Wicheon Town, Mt. Geumwon, 35°43′40″ N, 127°45′50″ E, 1351 m, on bark, June 25, 2010, X.Y. Wang et al. 100552. Jeju-do. Jeju-si: Aewol-eup, Nabeup-ri, subtropical forest, 33°26′05″ N, 126°19′49″ E, 106 m, on rock, July 5, 2012, S.O. Oh et al. 121494, 121833. Jeollabuk Province. Jangsu County, Jangsu Village, Daeseong Road, Mt. Palgongsan: 35°36′01″ N, 127°27′57″ E, 688 m, on bark, September 10, 2005, J.S Hur 050395, 050411, 050404, 050408. Jeollanam Province. Gurye County: Gwangui Town, Hanguktongsin Road, in the park of Sunchon University Education Centre, 34°14′56″ N, 127°28′24″ E, 90 m, on bark, June 26, 2015, S.Y. Kondratyuk et al. 150431.

##### *Phaeophyscia primaria* (Poelt) Trass.

Thallus foliose, greenish grey to grey greenish, loosely attached substrate, 5–13 cm in diameter; lobes irregularly or dichotomously branched, discrete or imbricate, concave, usually upturned near the tips, 2–8 mm wide; upper surface lacking soredia, isidia, and lobules; thallus 150–290 µm thick; upper cortex paraplectenchymatous, 20–45 µm thick; algal layer 30–50 µm thick, algae cells green, 5–14 µm in diameter; medulla white loose, 60–200 µm thick; lower surface black, usually becoming pale to brown toward lobe tips, rhizinate; rhizines dense, black, simple, 1–4 mm long; lower cortex paraplectenchymatous, blackish-brown, 22–35 µm thick; apothecia rare, up to 4 mm in diameter; margin entire or slightly crenate, lacking cortical hairs; disc reddish brown, flat to concave; amphithecium occasionally with retrosely hair near the base; hymenium 70–90 µm high; hypothecium hyaline or very pale straw-yellow, 40–55 µm thick; asci clavate, eight-spored, 65–75 × 14–18 µm; spores brown, *Physcia* type, 16.5–18.5 × 6.9–8.5 µm; pycnidia common, immersed into thallus, brown to black; conidia ellipsoid, 2.5–3 × 1–1.5 µm.

Chemistry: Thallus K−; medulla K−; lichen substance absent.

Ecology: The species is found growing on saxicolous mosses in a shaded environment.

Distribution: *Phaeophyscia primaria* is distributed widely in China, Nepal, Japan, and South Korea [4,6,14,45].

Comments: This species is characterized by distinctly concave, branched, and broad lobes (2–8 mm wide), which are usually upturned near the tips, lacking asexual propagules, a black lower surface with simple black rhizines, white medulla, apothecial margin without cortical hairs, *Physcia*-type spores, and absence of lichen substance.

This species resembles *Phaeophyscia ciliata*, and was also treated as *P. ciliata* in Japan for a long time until the revision on this species, and three species were isolated [6]. These are similar in having a greyish thallus without asexual propagules, but the lobes of the *P. primaria* are concave and much broader (3–7 mm wide) than those of *P. ciliata* (<1.5 mm wide). The present species was recorded in the Sorak Mountain and Chiri Mountain in South Korea [32,34] and extends to Nepal, China, and Japan [4,6]. The species usually grows on the moss over bark or rocks.

*Phaeophyscia primaria* is similar to *P. trichophora*, which was reported in Korea based on two specimens from Mt. Naejang (Hur 050006) and Baegdam Temple (Hur 041524) [60], but *P. primaria* can be distinguished from the latter species by the broader lobes without asexual propagules, apothecia lacking cortical hair around the margin and black lower surface, and these two specimens should be *P. primaria*.

Representative specimens examined: Chungcheongbuk Province. Jecheon-si, Hansu-myeon, Mt. Worak, 36°51’36" N, 128°05’27" E, 245 m, on saxicolous mosses, September 18, 2004, J.S. Hur 041160, 041164. Gangwon Province. Inje County, Buk Town, Baekdam Temple, 38°11’16" N, 128°21’42" E, 450 m, on saxicolous mosses, October 11, 2004, J.S. Hur 041507, 041524. Gyeonggi Province. Pocheon City, Idong Town, 38°06’24.8" N, 127°23’20.5" E, 535 m, on saxicolous mosses, July 28, 2008, J.S. Hur 080338; Gyeongsangbuk Province. Mungyeong City, Mungyeong Village, Mungyeongsaejae Provincial Park, 36°47’21.3" N, 128°04’26.3" E, 468 m, on moss over soil, January 1, 2017, D. Liu 170619, 170624. Jeju Province. Seogwipo City, Seongsan Village, Goseong Road, Seopjicoji, 33°19’21" N, 126°50’49" E, 69 m, on saxicolous mosses, June 19, 2014, S.Y. Kondratiuk 140328. Jeollabuk Province. Jeongeup City, Naejang District, Mt. Naejang, 35°29’44" N, 126°53’41" E, 535 m, on saxicolous mosses, January 8, 2005, J.S. Hur 050006.

##### *Phaeophyscia pyrrhophora* (Poelt) D.D. Awasthi and M. Joshi.

Thallus foliose, shiny gray to dark brown, tightly attached to the substrata, 3–10 cm in diameter; lobes branched irregularly, or occasionally elongate, contiguous, concave, lacking soredia, isidia, and lobules, 1–6 mm wide; thallus 100–240 µm thick; upper cortex paraplectenchymatous, 15–45 µm thick; algal layer 25–40 thick; alga cells green, 5–12 µm in diameter; medulla orange, loose, 80–200 µm thick; lower surface black, sometimes becoming pale brown to white at the apex, rhizinate; rhizines dense, black and becoming white or at least in the tip near the lobe apex, simple, up to 3 mm long; lower cortex paraplectenchymatous, blackish brown, 20–30 µm thick; apothecia up to 5 mm in diameter; margin entire or with marginal lobules, lacking cortical hairs; disc reddish brown, flat to con-cave; amphithecium sometimes with numerous black hairs, hymenium 80–110 µm high; hypothecium hyaline or very pale brown, 60–80 µm thick; asci 8-spored, 70–98 × 10–12 µm; spores dark brown when mature, *Pachysporaria* type, often with apical sporoblastidia, 21.5–30 × 8–13 µm; pycnidia common, brown to black; conidia ellipsoid, 2–3.5 × 1.5–2 µm.

Chemistry: Thallus K−; medulla K+ violet; skyrin present.

Ecology: The species is usually found growing on the bark of *Abies*, *Betula*, and *Quercus*; one specimen occurs on rocks in South Korea.

Distribution: *Phaeophyscia pyrrhophora* is distributed widely in China, Japan, South Korea, Nepal, and the eastern parts of Russia [4,14,18,33,46].

Comments: This species is characterized by broad, contiguous, or imbricate lobes (up to 5 mm in length), lacking asexual propagules, a black lower surface with a paraplectenchymatous lower cortex, apothecial margin without cortical hairs, spores of the *Pachysporaria* type, and containing skyrin, but lacking zeorine.

*Phaeophyscia pyrrophora* is similar to *Physcia erythrocardia* in having a similar thallus containing skyrin, but it can be distinguished from the latter species by a *Pachysporaria*-type spore rather than a *Physcia* type. In addition, this species lacks zeorin, which is constantly produced in *Physcia erythrocardia* [2,18]. This species differs from *P. endococcinodes* in having broader lobes, and is usually corticolous, whereas the latter is saxicolous and zeorine is occasionally present. Based on these points, the specimens recorded as *P. endococcinodes* [60] were found to be *P. pyrrophora* according to the morphological and chemical analyses. *Phaeophyscia pyrrophora* is apparently closely related to *P. fumosa* Moberg, a species reported in Kenya; however, the lobes of *P. pyrrophora* are concave and broader, whereas they are rather flat and narrower in *P. fumosa*.

Representative specimens examined: Gangwon Province. Taebaek City, Gumunso District, Mt. Hambaek: 37°11’12" N, 128°54’56" E, 1361 m, on bark, June 19, 2007, J.S Hur 070703. Sodo District, Mt. Taebaek, 1340 m, on rock, September 12, 2004, J.S. Hur 041083. Gyeongsangbuk Province. Bonghwa County, Mulya Town, Mt. Seondal, 37°02’13" N, 128°42’42" E, 1238 m, on bark, April 24, 2007, J.S. Hur 070263, 070268. Gyeongsangnam Province. Hamyang County, Macheon Town: Changwon Road, Mt. Baekun, 35°38’30" N, 127°37’13" E, 1071 m, on trunk, July 4, 2015, J.J. Woo et al. 150731, 150733, 150751, 150759. Jeju Province. Jeju City, Mt. Halla, 33°21’30" N, 126°30’14" E, 1633 m, on bark, November 9, 2014, S.O Oh et al. 141443, 141444; Yeongsil Trail, 35°18’16" N, 127°34’14" E, 1290 m, on bark, July 4, 2016, D. Liu 162409.

##### *Phaeophyscia rubropulchra* (Degel.) Moberg.

Thallus foliose, 1–3 cm in diameter; lobes dichotomously or irregularly branched, 0.5–1.5 mm wide; upper surface greenish gray or greenish brown, plane or more or less concave, epruinose, sorediate, soralia usually terminal or marginal, lip-shaped, occasionally laminal or capitate; thallus 110–200 µm thick; upper cortex paraplectenchymatous, 20–30 µm thick; algal layer continuous, about 25–40 µm thick, green alga, cells 7–14 µm in diameter; medulla orange or white with an orange band in the lower medulla, loose, 70–90 µm thick; lower cortex paraplectenchymatous, 25–30 µm thick; lower surface black or becoming pale to white toward the lobe tip, rhizinate; rhizines rather dense, black, occasionally pale to even white at tip, up to 2 mm long; apothecia not seen; pycnidia immersed into thallus, brown to black; conidia ellipsoid, 2–3.5 × 1.5–2.4 µm.

Chemistry: Thallus K−; medulla K+ violet; skyrin present.

Ecology: The species is found growing on the rocks and bark of *Acer* spp., *Cornus controversus*, *Fraxinus* sp., *Lindera obtusata*, *Meliosma myriantha*, *Cedrus deodara*, *Cerasus*, and *Pinus densiflora*.

Distribution: *Phaeophyscia rubropulchra* is distributed widely in Asia, North America, Russia, and East Asia [2,4,7,41].

Comments: *Phaeophyscia rubropulchra* is characterized by well-developed lobes with terminal or marginal soralia, and orange-red medulla. *Phaeophyscia rubropulchra* is apparently closely related to *P. endococcina* and *P. erythrocardia* from which it can be distinguished by the presence of soredia. The species can be saxicolous and corticolous in Korea, when it occurs in a saxicolous and shading environment; the thallus is much darker and thicker; the orange color of the medulla is also much more obvious; the lower surface is almost black even toward to the lobe tips, whereas the thallus is usually gray to pale greenish, the medulla light orange, or even the center and lower part of medulla orange; that of the tip and upper parts are white, the lower surface is usually brown to black, or white at the lobe tip occasionally if found in the corticolous and exposed surroundings.

Representative specimens examined: Gangwon Province. Jeongseong County, Gangneung City, tourist pass toward peak Seokbyeong Mt., 37°34’39" N, 128°51’24" E, 760 m, on bark, July 10, 2015, S.Y. Kondratyuk and L. Lőkös 150954. Yeongwol County, Sangdong Village, Gurae Road, nearby Manhang-jae, 37°9’21" N, 128°53’28" E, 1260 m, on bark, July 17, 2014, U. Jayalal et al. 141303, 141304, 141329. Gyeongsangnam Province. Geochang County, Seolcheon Town, Mt. Sambong, 35°52’07" N, 127°49’42" E, 970 m, on rock, Jun 18, 2015, S.O. Oh et al. 150014, 150020. Jeju Province. Jeju City, Temple Gwanum, 33°25’22" N, 126°33’35" E, 615 m, on rock, July 7, 2012, S.Y. Kondratyuk and L. Lőkös 121921, 121926, 121930. Seogwipo City, Namwon Village, Mt. Halla, Yeongsil Trail, 33°21’11" N, 126°29’48" E, 1290 m, on rock, July 4, 2016, D. Liu 162431. Jeollabuk-do, Muju County, Mupung Town, Mt. Daedeok, 35°54’36" N, 127°52’57" E, 981 m, on rock, June 19, 2015, S.O. Oh et al. 150132. Jeollanam Province. Jangheung County, Gwansan Village, Okdang Road, Cheongwansan Mts, near the entrance, 34°32’55" N, 126°55’60" E, 91 m, on bark, June 23, 2015, S.Y. Kondratyuk and L. Lőkös 150425.

##### *Phaeophyscia spinellosa* Kashiw.

Thallus foliose, whitish grey, loosely attached substrate, 5–10 cm in diameter; lobes irregularly or dichotomously branched, discrete or imbricate, concave, 1–2 mm wide; thallus 100–200 µm thick; upper surface soredia, isidia and lobules absent, slightly pruinose sometimes; upper cortex paraplectenchymatous, 30–45 µm thick; algal layer 30–55 µm thick, algae cells green, 5–14 µm in diameter; medulla white, loose, 60–120 µm thick; lower surface black, usually becoming pale to brown toward the lobe tips, rhizinate; rhizines dense, black, simple, 1–4 mm long; lower cortex paraplectenchymatous, blackish brown, 20–30 µm thick; apothecia numerous if present, up to 4 mm in diameter; margin entire, cortical hair sparsely present on apothecia margin; disc reddish brown, flat to concave; amphithecium occasionally with retrosely hair near the base; hymenium 50–65 µm high; hypothecium hyaline or very pale straw-yellow, 40–50 µm thick; asci clavate, 8-spored, 56–73 × 10–13 µm; spores brown, *Physcia* type, 16.4–*19.4*–23 × 7.3–*8.9*–11.4 µm, Q = 2.0–2.6, Q(m) = 2.2, *n* = 65; pycnidia common, immersed into thallus, black; conidia ellipsoid, 2.5–3 × 1–1.5 µm.

Chemistry: Thallus K−; medulla K−; lichen substance absent.

Ecology: The species is found growing on saxicolous mosses.

Distribution: *Phaeophyscia spinellosa* is distributed widely in Japan and South Korea [6,30].

Comments: *Phaeophysica spinellosa* is characterized by a whitish grey thallus without asexual propagules, a black lower surface with black rhizines, white medulla, apothecia with sparse cortical hairs on the margin, spores *Physcia* type, and an absence of lichen substances.

This species is similar to *Phaeophyscia ciliata* (Hoffm.) Moberg, but it can be separated clearly from *P. ciliata* by the broader and imbracate lobes and the presence of cortical hairs on the apothecial margin; *Phaeophysica spinellosa* resembles *P. hirtuosa*, an Eastern Asiatic species, both of them lack asexual propagules, have apothecia with cortical hairs on the margins, and produce spores of the *Physcia* type. In addition, *P. spinellosa* differs from *P. hirtuosa* in the much sparser hair around the apothecia and usually grows on rocks, whereas *P. hirtuosa* usually has very dense hairs on the apothecial margin and grows on bark in the mountains in Eastern Asia.

Representative specimens examined: Jeju Province. Jeju City: Hallim Village, Gwideok Road, coast near Chorok Village, 33°26’33" N, 126°17’00" E, 18 m, on rock, July 5, 2012, S.Y. Kondratyuk et al, 121370; Hangyeong Town, Sinchang Road, nearby the coast around Singaemul Park, 33°30’31" N, 126°10’13" E, 19 m, on rock, July 5, 2012, S.Y. Kondratyuk et al, 121413, 121415, 121416. Jeollanam Province. Goheung County, Yeongnam Town, Ucheon Road, Yongam Village, Yongbawi seaside, 34°35’41" N, 127°30’19" E, 10 m, on rock, February 19, 2010, Y. Joshi et al. 100337, 100193. Jindo County, Jeob Island, 34°23’41" N, 126°18’9" E, 1 m, on rock, June 3, 2011, X.Y. Wang and J.A. Ryu, 110504, 110510, 110514, 110632. Wando County, Cheongsan Island, 34°09’10" N, 126°52’50" E, 2 m, on rock, June 23, 2011, X.Y. Wang and J.A. Ryu 110727, 110741.

##### *Phaeophyscia squarrosa* Kashiw.

Thallus foliose, greenish gray to grayish brown, lobate, 2–10 cm in diameter; lobes dichotomously or irregularly branched, contiguous, lobulate, 0.5–3(–4) mm wide, lobules marginal or submarginal, laminal occasionally, dorsiventral, numerous, 0.2–0.5 mm wide; upper surface more or less convex, epruinose; medulla white; thallus 130–200 µm thick; upper cortex paraplectenchymatous, composed of 4–5 layers of cells, 25–40 µm thick; algal layer continuous, approximately 30 µm thick, algae cells 7–17 µm in diameter; medulla white, loose, 75–100 µm thick; lower surface black at the center of lobes and pale brown or even white near the periphery, rhizinate; rhizines dense, black, becoming white toward tips or white near the lobe periphery, simple near lobe periphery or squarrosely branched, up to 6 mm long; lower cortex paraplectenchymatous, dark brown or pale brown near the periphery of lobes, composed of 2–3 layers of inflated cells, 25–40 µm thick. Apothecia rare, cup-shaped, approximately 2.5 mm in diameter; margin entire; disc reddish brown, epruinose; amphithecium retrorsely hairy at the base; hymenium 70–80 µm high; hypothecium pale brown, 30–40 µm thick; asci clavate, eight-spored, 64–76 × 12–17 µm; spores brown, *Physcia* type, 18.6–*21.7*–26 × 8.1–*10*–12.6 µm, Q = 1.7–2.6, Q(m) = 2.15, *n* = 73. Pycnidia immersed in thalli, black spotted; conidia 2.5–3 × 1–1.5 µm.

Chemistry: Thallus K−; medulla K−, C−, KC−, P−; contain zeorine.

Ecology: This species is found frequently growing on the bark of *Quercus* spp. or sometimes on moss over bark.

Distribution: *Phaeophyscia squarrosa* is distributed in North America, Japan, and South Korea [17,60].

Comments: This species is characterized by broad lobes with numerous dorsiventral lobules near the margin, white medulla, brown to black lower surface of lobes covered with squarrosely branched rhizines together with simple ones, cup shaped apothecia with retrorsely hairy amphithecium at the base, *Physcia*-type spores, and contain zeorine as constant substances in the thallus.

The present species is closely related to *P. ciliata* and *P. hispidula,* from which it can be distinguished by the presence of lobules; *Phaeophyscia squarrosa* is most similar to *P. exornatula*, which has a wide distribution in Asia and lobules, but the lobules of the latter species come from submarginal pustules that usually break into soralia. In addition, *P. exornatula* lacks zeorin and can grow on rock or saxicolous mosses, whereas *P. squarrosa* has zeorine and grows frequently on the bark; this species may be confused with *P. laciniata* Mull. Arg., a species known from Costa Rica and North America; all of them have lobules, but it can be distinguished from the latter species by white medulla due to lacking skyrin. This species might be confused with *P. primaria* because they both have larger thalli without asexual propagules and share a similar habitat, but they differ in having squarrosely branched rhizines and constantly containing zeorine.

*Phaeophyscia imbricata* (Vain.) Essl., is a species recorded in Japan and North America [2,17,19,60], and recorded as containing zeorine with lobes, but the type specimen examined by Moberg [4] is not well-developed and does not contain zeorine. Consequently, the species was treated as *P. imbricata* (Vain.) Essl. as a variation of *P. hispidula*. Some lichenologists have used erect lobules as the identical characteristics for *P. imbricata* [2,60], but these erected lobules are also quite common in *P. exornatula*, which lacks zeorine. Eventually, all of the specimens cited by Wei & Hur. [60] and regarded as *P. imbricata* were placed into *P. squarrosa* in this study.

Representative specimens examined: Chungcheongbuk Province. Yeongdong-gun, Sangchon-myeon, Mulhan-ri, Mt. Samdobong, 36°01’14" N, 127°52’41" E, 982 m, on trunk, July 1, 2015, J.J. Woo et al. 150435. Gangwon-Do. Hongcheon County, Nae Town, Mt. Eungbok, 37°51.772’ N, 128°33.528’ E, 866 m, on *Quercus*, May 23, 2009, Y. Joshi et al. 090745. Gyeongsangbuk Province. Mungyeong City, Sanbuk Town, Mt. Gongdeok, 36°44’42" N, 128°15’54" E, on bark, June 20, 2007, J.S. Hur 070774. Gyeongsangnam Province. Hamyang-gun, Seosang-myeon, Mt. Baekun, 35°36’18" N, 127°38’26" E, 903 m, on bark, August 17, 2006, J.S. Hur 060586. Jeollanam Province, Gurye County, Toji Town, Mt. Jiri, 35°19’26" N, 127°36’46" E, 1555 m, on *Quercus*, September 28, 2006, J.S Hur 060864.

##### *Physciella melanchra* (Hue) Essl.

Thallus foliose, greenish gray, pale ochraceous buff in herbaria, 1–4 cm in diameter; lobes irregularly branched, linear-elongate or more or less broadened towards the apices, rounded, irregular or flabellate, 0.5–1.0(–1.5) mm wide; thallus 100–170 µm thick; upper surface plane or more or less concave, epruinose, sorediate, soralia laminal, punctiform (0.3–1 mm in diameter), often spread over lobes; upper cortex paraplectenchymatous, composed of 2–3(–5) layers of cells (3–5 µm in diameter), 15–50 µm thick; algal layer continuous, approximately 40–60 µm thick, algae 5–12 µm in diameter; medulla white, loose, 40–80 µm thick; lower surface white to pale tan, rhizinate; rhizines sparse, usually concolorous with a lower surface; lower cortex colorless, prosoplectenchymatous, 25–50 µm thick. Apothecia lecanorine type, frequent in large specimens but rare in small specimens, cup-shaped, 1–2 mm in diameter; disc reddish brown when immature, turns dark brownish to black after maturity, epruinose; margins entire or slightly crenulate, occasionally with punctiform soralia; proper exciple 25–30 µm thick; hymenium 70–100 µm high; hypothecium colorless, 30–45 µm thick; asci clavate to cylindrical, 8-spored, 60–65×9.1–12 µm; spores *Physcia* type, brown, 17–24 × 7.1–10.5 µm. Pycnidia immersed in the thallus, black-spotted, conidia about 2.5–3.5 × 1–1.5 µm (Figure 4).

Chemistry: Thallus K−; medulla K−, C−, KC−, P−; lacking secondary substances.

Ecology: The species is found growing on rocks in South Korea, together with *Phaeophyscia primaria*, *P. orbicularis*, and *P. exornatula*.

Distribution: This species has a wide distribution in North America, Asia, Europe, and South Korea.

Comments: *Physciella melanchra* is characterized by a gray thallus lacking atranorin and with laminal, punctiform soralia, a prosoplectenchymatous lower cortex, and white to tan lower surface. *Physciella melanchra* is similar to *Hyperphyscia adglutinata* by having soralia, pale lower surface, and prosoplectenchymatous lower cortex, but differs in having ellipsoid conidia; *Physciella melanchra* can be distinguished from *P. chloantha* by laminal soralia, whereas *P. chloantha* has marginal or terminal soralia.

Representative specimens examined: Gyeongnam. Namhae, Mt. Mangeun, 34°51’4" N, 127°49’36" E, 160 m, on rock, April 28, 2011, X.Y. Wang and J.A. Ryu 110153, 110157, 110121. Jeju, Aewol Village, Nabeup Road, subtropical forest, 33°26’06" N, 126°19’48" E, 90 m, on rock, July 5, 2012, S.O. Oh et al. 121829, 121833; Jeollabuk. Gochang, Asan, Mt. Seonun: 50 m, on rock, May 11, 2003, J.S. Hur 030226. Jeollanam, Suncheon: Along the Dongcheon River, 34°57’46" N, 127°29’17" E, on bark, October 2, 2011, S.Y. Kondratyuk 110979; Sunchon National University: 34°58’01" N, 127°28’47" E, 47 m, on *Cerasus* sp., December 1, 2017, D. Liu 171448, 171449, 171450; humanitarian faculty, 34°58’10" N, 127°28’36" E, on bark, October 4, 2011, S.Y. Kondratyuk 110985, 110992; 34°58’12" N, 127°28’57" E, 28 m, on stairs, June 8, 2017, D. Liu 170625; Seokhyeon District, Hyanglim, 34°58’20" N, 127°28’43" E, 46 m, on rock, October 4, 2017, D. Liu 171426.

## 4. Discussion

### 4.1. Phylogeny Analysis

The foliose genera within the family Physciaceae were traditionally distinguished by their thallus, cortical structure, ascospores, and chemical substances [1,2,19,21,61,62]. Nordin et al. [63] selected parts of representative genera including the fruticose, crustose, and foliose groups, and combined the molecular and morphological data. Their results suggested that the foliose groups were gathered together, but species of *Physciella* were not included. We supplemented the ITS sequences of several species in *Physciella* with *Heterodermia*, *Hyperphyscia*, *Physcia*, *Physconia*, and *Phaeophyscia*, and then four characteristics (atranorin, ascospores type, and upper and lower cortex type) were used to examine the relationship among species. Seven genera separated from each other and formed a single clade. All were monophyletic. The genus *Physciella* is strongly supported as being a sister to *Phaeophyscia* and can be distinguished from the latter by having a prosoplectenchymatous lower cortex, which also provides support for separating these two genera based on the difference in the lower cortex [4,20]. Morberg [20] combined the species *Physcia poeltii* Frey into *Phaeophyscia* as *Phaeophyscia poeltii* (Frey) Nimis based on having a white lower surface and a prosoplectenchymatous lower cortex. According to our phylogenetic analysis, *Phaeophyscia poeltii* is in the *Physciella* clade, and by combining the cortex characters from Morberg’s identification [20], we propose treating the new combination as *Physciella poeltii* (Frey) D. Liu and J.S. Hur.

Korean specimens lacking atranorin and with *Physcia-* or *Pachysporaria*-type ascospores belong to three clades: *Phaeophyscia*, *Physciella*, and *Hyperphyscia*; the squamulose species with light orange medulla, *Hyperphyscia crocata* is close to *Hyperphyscia confusa*. This species shares the same type of lower cortex and conidia with *H. confusa* and *Hyperphyscia adglutinata*. In contrast, the specimens with ellipsoid conidia and a prosoplectenchymatous lower cortex formed a clade with the type species *Physciella chloantha*, which was not well separated from *P. melanchra* in terms of laminal soralia. The genus was divided into two large clades without support, and the species had orange medulla mixed with white medulla; however, more information was difficult to obtain due to the lesser support. *Phaeophyscia squarrosa* and *Phaeophyscia constipata* had white medulla, but *P. pyrrhophora*, *P. encoccinodes*, *P. endococcina*, and *P. rubropulchra* had orange medulla.

### 4.2. Taxonomy Revision of Phaeophyscia, Hyperphyscia and Physciella in South Korea

The *Phaeophyscia* species was first reported in South Korea [30] with two species, *Physcia endococcina* (= *Phaeophyscia endococcina*) and *P. lithotea* var. *sciastra* (*Phaeophyscia sciastra*), whereas Korean specimens of *Phaeophyscia endococcina* were re-identified as *P. endococcinodes* [42,60]. Previous studies have reported that the *Phaeophyscia encoccinodes* specimens were *P. pyrrhophora* based on a re-examination from the morphology and TLC results; *P. encoccinodes* was saxicolous and had smaller lobes [2,18], and specimens of *P. erythrocardia* were assigned to two different species: *P. pyrrhophora* and *P. endococcinodes*. These two species can be distinguished from *P. erythrocardia* by *Pachysporaria*-type spores, whereas *P. erythrocardia* has *Physcia*-type spores, zeorine is constantly present, and there are smaller amphithecium cortical cells [2,18]. As a result, *Phaeophyscia endococcina* and *P. erythrocardia* do not occur in South Korea. *Phaeophyscia trichphora* is characterized by *Pachysporaria*-type spores, whereas the specimens cited in Wei & Hur [60] have *Physcia*-type spores; all are now referred to as *P. primaria*.

The genera *Hyperphyscia* and *Physciella* share a similar lower cortex, prosoplectenchymatous if present, but *Hyperphyscia* can be distinguished from *Physciella* by the filiform conidia, whereas those of the latter species are ellipsoid. *Hyperphyscia adglutinata* is characterized by a small thallus with an indistinctly prosoplectenchymatous lower cortex and filiform conidia, and was reported in South Korea [43]. All the specimens cited are now referred to as *Physciella melanchra* based on the morphology and molecular data, and *Hyperphyscia adglutinata* should be excluded from the lichen flora of South Korea. The morphology of *Physciella melanchra* varies considerably in different substrates. The thallus is often orbicular on the rock, but is frequently irregular and smaller when found on bark.

        **Worldwide key to species of *Phaeophyscia* and *Physciella***

1a. Medulla orange ………………………………………… 2

1b. Medulla white ………………………………………… 10

2a. Soredia present ………………………………………… 3

2b. Soredia absent ………………………………………… 4

3a. Lobe narrow, always <1.5 mm, rhizine under the apothecia numerous, zeorine present, corticolous or saxicolous ………… ***P. rubropulchra***

3b. Lobe always up to 2 mm, rhizine under the apothecia rarer, zeorine absent, saxicolous ………… *P. endophoenicea*

4a. Lobules present ………………………………………… 5

4b. Lobules absent ………………………………………… 7

5a. Upper surface and apothecia with cortical hair, thallus size always <2 cm, corticolous ……………………… ***P. hunana***

5b. Upper surface and apothecia without cortical hair, size >2 cm, saxicolous ……………………… 6

6a. Lobe 1–2 mm wide, lobules numerous, zeorine absent ……………………… *P. laciniata*

6b. Lobe <1 mm, lobules rare, zeorine occasionally present ……………………… ***P. endococcinodes***

7a. Spore *Pachysporaria*-type, zeorine absent ………………………………………… 8

7b. Spore *Physcia*-type, zeorine present ………………………………………… 9

8a. Lobe broad, >1 mm, apothecia with numerous rhizines ……………………… ***P. pyrrhophora***

8b. Lobes narrow, <1 mm, apothecia without rhizines ……………………… *P. fumosa*

9a. Amphithecial cortex of smaller cells, 3–7(10) μm, lobes flat, ca. 1 mm broad, on bark, saxicolous mosses ………………… *P. erythrocardia*

9b. Amphithecial cortex of larger cells, 6–12(15) μm, lobes flat to weakly convex, ca. 0.5 mm broad, saxicolous ………………… ***P. endococcina***

10a. Thallus lower surface dark brown to black, with numerous black rhizines ………………………………………… 11

10b. Thallus lower surface white to pale tan to pale brown, never black, with numerous concolorous rhizines ………………… 29

11a. Lobules, isidia, or soredia absent ………………………………………… 12

11b. Lobules, isidia, or soredia present ………………………………………… 16

12a. Lower surface pale brown to dark brown ………………… ***P.***
***esslingeri***

12b. Lower surface black, rhizine simple, spore *Physcia*-type ………………………………………… 13

13a. Cortical hairs present on lobe end, lobe narrower 0.5–1(1.5) mm broad ………………… *P. hirtella*

13b. Cortical hairs absent on the lobe end, lobe broader, >1 mm broad ………………………………………… 14

14a. Lobe broader, >3 mm, cortical hair absent on apothecia margin ………………… ***P. primaria***

14b. Lobe narrower, <3 mm, cortical hair on apothecia margin more or less present ………………………………………… 15

15a. Cortical hair sparse on apothecia margin, spore size smaller, 15–18 × 7–11, on rock ………………… ***P. spinellosa***

15b. Cortical hair numerous on apothecia margin, spore size larger, 17–23 × 9–11, on bark ………………… ***P. hirtuosa***

16a. Upper surface or lobes end more or less pruinose ………………………………………… 17

16b. Upper surface or lobe without pruina ………………………………………… 21

17a. Isidia or soredia present ………………………………………… 18

17b. Isidia or soredia absent ………………………………………… 19

18a. Upper surface dark brown to black, without cortical hair, isidia dark to black ………………… ***P. sciastra***

18a. Upper surface gray brown to dark brown, with cortical hair, isidia usually developed into ciliated lobes ………………… *P. kairamoi*

19a. Zeorine absent ………………………………………… *P. ciliata*

19b. Zeorine present ………………………………………… 20

20a. Lobe narrower, 0.2–0.5 mm, lobules sparse or absent, on rock …………………… *P. decolor*

20b. Lobe broader, 0.5–3(5) mm, lobules numerous, saxicolous or corticolous …………………… ***P. squarrosa***

21a. Lobules present ………………………………………… 22

21b. Lobules absent ………………………………………… 24

22a. Lobules on the lobe tips, soredia present …………………… ***P. adiastola***

22b. Lobules on the lobe submargin or margin, soredia absent ………………………………………… 23

23a. Lobe narrower, 0.2–0.5 mm, radiating, lobules on the margin, apothecia margin with cortical hair, *Pachsporaria*-type spora …………… ***P. confusa***

23b. Lobe broader, 1.5–3 mm, branched, lobules on the submargin, apothecia margin without cortical hair, *Physcia*-type spora …………… ***P. exornatula***

24a. Isidia laminal or central ………………………………………… *P. crocea*

24b. Isidia absent ………………………………………… 25

25a. Lobe narrower, <2 mm ………………………………………… 26

25b. Lobe broader, >2 mm ………………………………………… 28

26a. Cortical hair present on the lobe end and apothecia margin, soralia lip-shaped ………………………… *P. hirsuta*

26b. Cortical hair absent, soredia powdery to granular ………………………………………… 27

27a. Soredia primary on the lob ends then strongly labriform or capitate ………………………… *P. pusilloides*

27b. Soredia marginal or submarginal not strongly labriform or capitate ………………………… ***P. orbicularis***

28a. Soralia elongate, common on rock or bark in Asia ………………………… ***P. limbata***

28b. Soralia spherical to capitate, mainly on bark in Asia ………………………… ***P. hispidula***

29a. Lower cortex paraplectenchymatous ………………………………………… 30

29b. Lower cortex prosoplectenchymatous ………………………………………… 39

30a. Isidia present ………………………………………… *P. lygaea*

30b. Isidia absent ………………………………………… 31

31a. Cortical hairs present lobes tips ………………………………………… 32

31b. Cortical hairs absent ………………………………………… 34

32a. Conidia cylindrical ………………………………………… *P. culbersonii*

32b. Conidia ellipsoid ………………………………………… 33

33a. Upper surface pruinose, white spotted, lobe broader, 0.8–1 mm, apothecia margin with cortical hair ………………………… *P. cernohorskyi*

33b. Upper surface not pruinose, lobe narrower, 0.2–0.5 mm, apothecia margin without cortical hair ………………………… *P. nigricans*

34a. Soredia present ………………………………………… 35

34b. Soredia absent ………………………………………… 36

35a. Lobe 0.2–0.5 mm broad, thallus size smaller, <2 cm ………………………… *P. insignis*

35b. Lobes 0.5–1 mm broad, thallus size larger, 2–4(5) cm ………………………… *P. nashii*

36a. Lower surface rhizines numerous, rhizines present under apothecia ………………………………………… 37

36b. Lower surface rhizines rather sparse, rhizines absent under apothecia ………………………………………… 38

37a. Upper surface slightly rocky with maculae ………………………………………… *P. leana*

37a. Upper surface without maculae, apothecia sessile, apothecia with numerous rhizines ………………………… ***P.***
***trichophora***

38a. Thallus more or less orbicular, growing on bark or rock ………………………… *P. sonorae*

38b. Thallus irregular and tangle to sub-fruticose, growing on soil or moss ………………………… *P. constipata*

39a. Soredia present ………………………………………… 40

39b. Soredia absent ………………………………………… 41

40a. Soralia primarily terminal and marginal, labriform (lip-shape) ………………………… *Physciella chloantha*

40b. Soralia laminal and submarginal, rounded or irregular ………………………… ***Physciella melanchra***

41a. Spore *Physcia*/*Pachysporaria*-type, broader, 22–27 × 9–11.5 μm, and upper surface occasionally lobules present in the central ***Physciella denigrata***

41b. Spore *Physcia*-type, narrower, 17–23.5 × 6.5–9 μm, upper surface without lobules ………………… *Physciella nepalensis*

        Species in bold were recorded in Korea.

## Figures and Tables

**Figure 1 microorganisms-07-00242-f001:**
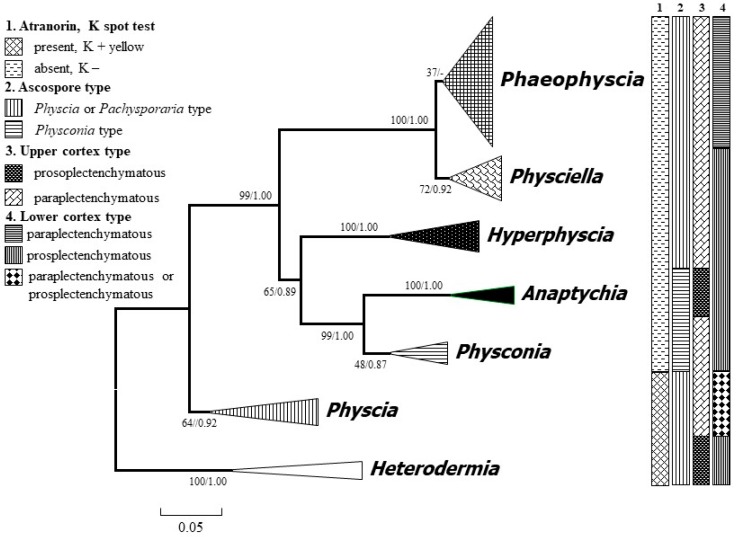
Consensus tree showing the phylogenetic relationships of the foliose major genera of the family Physciaceae based on the ITS (Internal Transcribed Spacer) sequence by applying maximum likelihood (ML) and Bayesian inference (BI). ML bootstrap (before the slash) and posterior probabilities from the Bayesian analysis are provided adjacent to the nodes. The species of each genus are concentrated as a single clade. *Heterodermia* was set as a root group.

**Figure 2 microorganisms-07-00242-f002:**
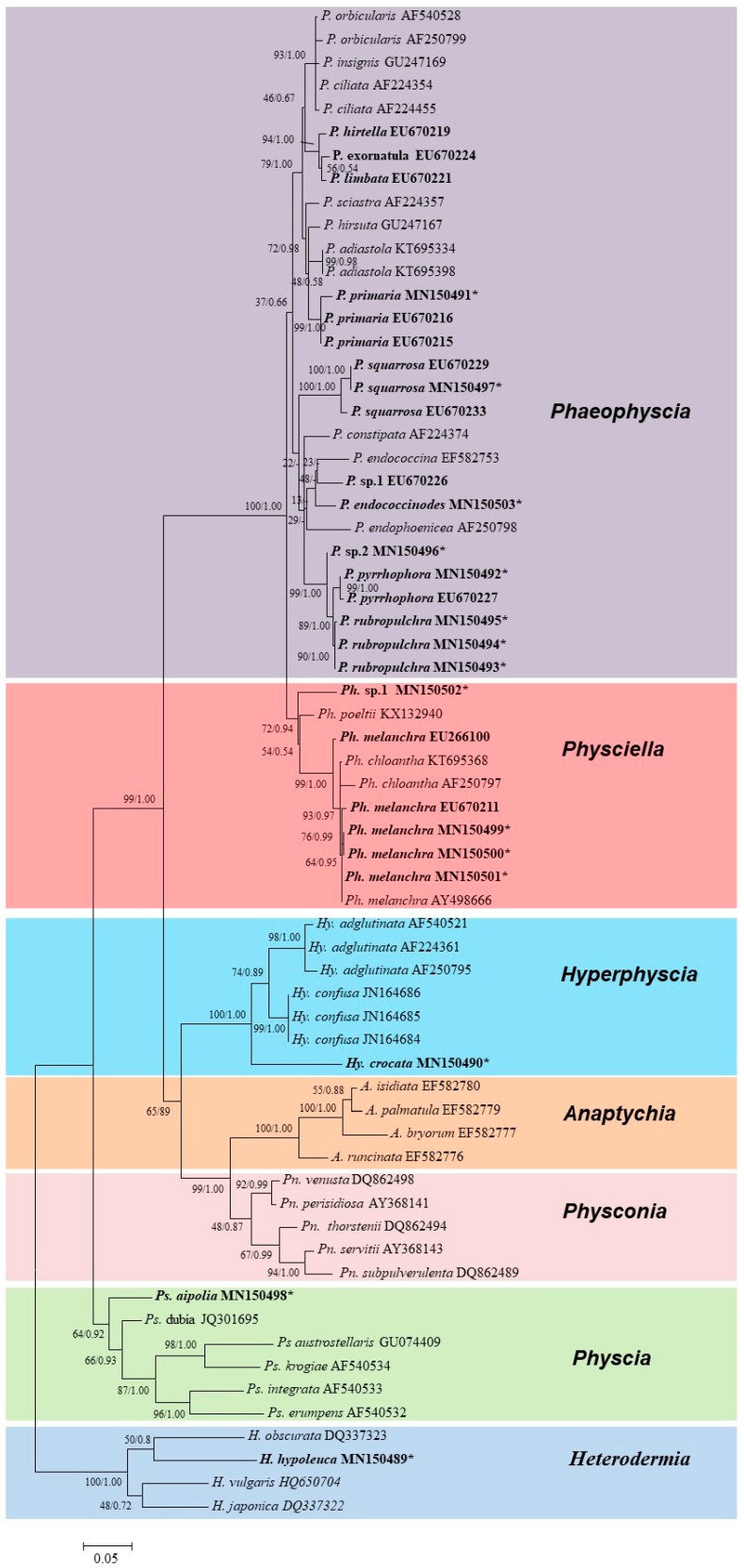
Consensus tree showing the phylogenetic relationships among the species in the family Physciaceae based on ITS sequence by ML and Bayesian inference. The related species of *Phaeophyscia*, *Physciella*, and *Hyperphyscia* are depicted. The ML bootstrap (before the slash) and posterior probabilities are given adjacent to the nodes. The phylogenetic tree was rooted by the species *Heterodermia*. Specimens from South Korea are in bold; the asterisk refers to the newly generated sequences in this study. *A.* = *Anaptychia*, *Hy.* = *Hyperphyscia*, *P.* = *Phaeophyscia*, *Ph.* = *Physciella*, *Pn.* = *Physconia*, and *Ps.* = *Physcia.*

**Figure 3 microorganisms-07-00242-f003:**
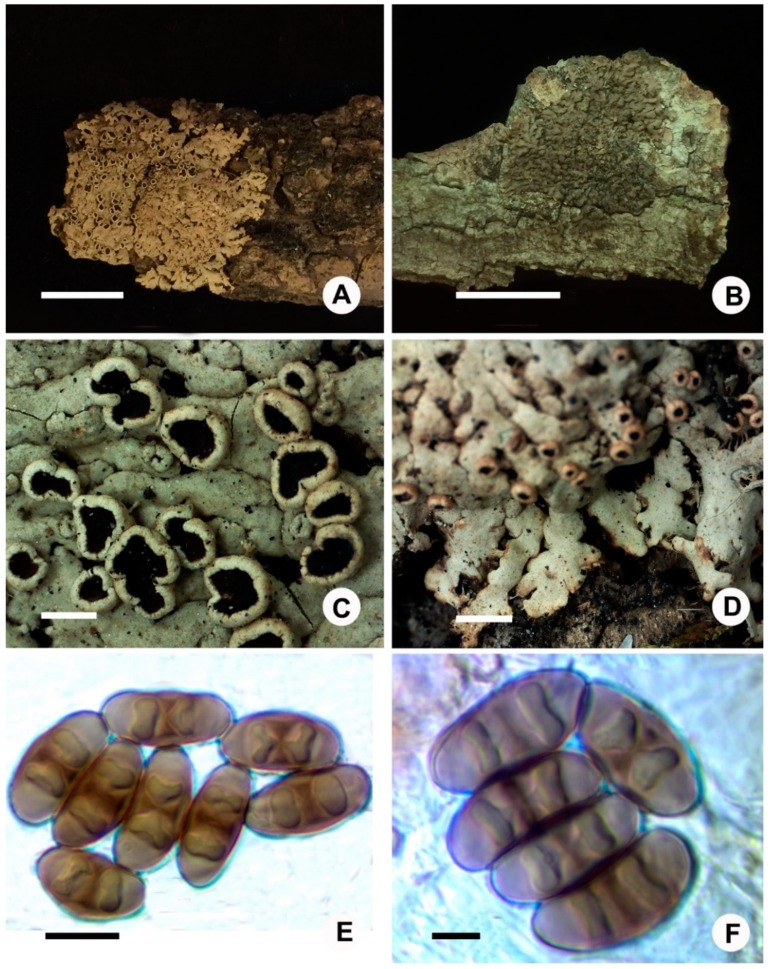
New records: (**A**,**C**–**E**), *Phaeophyscia leana* (J.S. Hur 040554), (**A**) habitat, (**B**) *Phaeophyscia hunana* (121894), (**C**) apothecia, (**D**) lobes, and (**E**,**F**) ascospores in oil. Scale bars: A,B = 1 cm, C,D = 1 mm, E = 10 µm, F = 5 µm.

**Figure 4 microorganisms-07-00242-f004:**
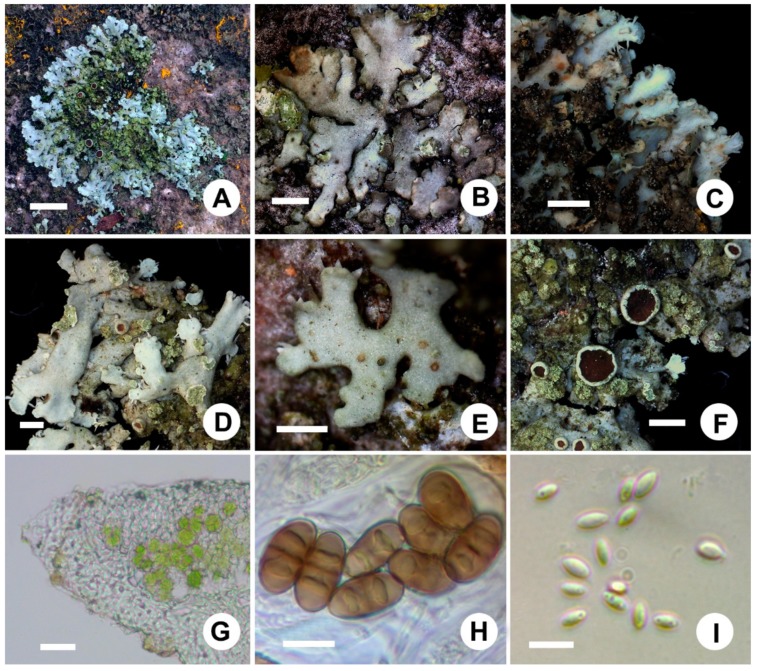
*Physciella melanchra* (**A**) on rock (D. Liu 171426), (**B**) on bark (D. Liu 171449), (**C**) white lower surface, (**D**) soralia and rhizines, (**E**) pycnidia, (**F**) apothecia, (**G**) vertical section of thallus, (**H**) ascospores in oil, and (**I**) conidia. Scale bars: (A) = 1 cm, (B) = 0.5 mm, (C) = 1 mm, (D) = 0.5 mm, (E) = 0.3 mm, (F) = 1 mm, (G) = 20 µm, (H) = 10 µm, (I) = 5 µm.

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
