# Peer review of "Revision of the Lichen Genus *Phaeophyscia* and Allied Atranorin Absent Taxa (Physciaceae) in South Korea"

_microorganisms, 2019, doi:10.3390/microorganisms7080242_

Round 1

Reviewer 1 Report

English can be improved, for example in the use of articles and other grammatical aspects.

Since Microorganisms is not a journal specialized in lichens, it would be good if the authors briefly introduced lichens in general and then began to describe the genus Phaeophyscia in particular.

At the end of the introduction, the authors should describe the objective of the work.

Table S1 with the published sequences of Physciaceae from GenBank is not in the supplementary material. In fact, as supplementary material, there is only the information about the vouchers of the specimens examined. I take the opportunity to mention that this information should be more systematized because, as it is presented, it is very cumbersome to review. In addition, the names of the species should be in italics.

On the other hand, the authors should make available in databases such as GenBank the sequences they obtained in this work, indicating in the manuscript the corresponding access numbers.

Physcia is not mentioned in the description of the phylogenetic tree of Figure 1. Likewise, the morphological characteristics encoded in frames and colours are not described in the text.

Again, the information in figure 2 is not sufficiently described in the text, for example, how many species make up each clade, why there are bold species in the tree, etc.

The assertions mentioned in the paragraph of lines 112 to 118 are not explained. For example, on what characteristics are the new descriptions of species based? At least it must be indicated that the features will be mentioned later. However, it must be mentioned in what characteristics the authors were based to propose the new classifications. Phaeophyscia koreana is not even in the phylogenetic tree. Classifications in species should not be based solely on morphological characteristics.

The discussion is rather a recapitulation of the results. The authors should discuss in more detail the proposed taxonomic changes related to phylogenetic clades. In addition, this being a journal about microorganisms in general, the authors should discuss that lichens are a multispecies symbiosis (Aschenbrenner et al., 2016 10.3389/fmicb.2016.00180) and state the difficulties that this particularity entails for the classification of these organisms (Spribille 2018 10.1016/j.pbi.2018.02.007).

Author Response

We very appreciated the editor and review’s comments to improve our manuscript.

We have revised our MS according to their comments and give our answer as following:

We marked corrections based on reviewers’ comment with yellow color.

We also marked corrections suggested from MDPI English-editing service with blue color.

Comment: English can be improved, for example in the use of articles and other grammatical aspects.

Answer: Professional English-editing service was conducted by MPDI (The edited MS can be downloaded from author submission page in MPDI system)

Comment: Since Microorganisms is not a journal specialized in lichens, it would be good if the authors briefly introduced lichens in general and then began to describe the genus Phaeophyscia in particular.

Answer: We added them in L27-29.

Lichens are one of the most successful symbiotes in nature, forming symbiotic associations with a fungus and algal partner, like green algae and cyanobacteria. In general, lichens can be divided into fruticose, foliose, and crustose lichens based on the growth type.

Comment: At the end of the introduction, the authors should describe the objective of the work.

Answer: We added it in L58-61.

In this study, we combined morphology, chemistry, and phylogenetic analysis with the aim of clarifying the relationship among the foliose genera in the Physciaceae family, and to investigate the constitution, distribution, and ecology of the atranorin-absent groups in Physciaceae more comprehensively.

Comment: Table S1 with the published sequences of Physciaceae from GenBank is not in the supplementary material. In fact, as supplementary material, there is only the information about the vouchers of the specimens examined. I take the opportunity to mention that this information should be more systematized because, as it is presented, it is very cumbersome to review. In addition, the names of the species should be in italics.

Answer: We additionally provided List S1 (full information of Korean specimens examined for the present study) with Table S1, and the species name was presented in italics.

Comment: On the other hand, the authors should make available in databases such as GenBank the sequences they obtained in this work, indicating in the manuscript the corresponding access numbers.

Answer: The Genbank accession numbers for the newly generated specimen were added into Table S1 with yellow color.

Comment: Physcia is not mentioned in the description of the phylogenetic tree of Figure 1. Likewise, the morphological characteristics encoded in frames and colours are not described in the text.

Answer: we have revised the paragraph and add more information of the genus Physcia, see line 1051062.

All of the atranorin-absent groups with K– formed a big clade and are separate from the atranorin groups (Heterodermia and Physcia) with K+.

Comment: Again, the information in figure 2 is not sufficiently described in the text, for example, how many species make up each clade, why there are bold species in the tree, etc.

Answer: we give more information for the topology of figure 2, see line 116–118.

Seven genera with 42 species were included in the topology, which contained Phaeophyscia (17 sp.), Physciella (3 sp.), Hyperphyscia (3 sp.), Anaptychia (4 sp.), Physconia (5 sp.), Physcia (6 sp.), and Heterodermia (4 sp.).

Comment: The assertions mentioned in the paragraph of lines 112 to 118 are not explained. For example, on what characteristics are the new descriptions of species based? At least it must be indicated that the features will be mentioned later. However, it must be mentioned in what characteristics the authors were based to propose the new classifications. Phaeophyscia koreana is not even in the phylogenetic tree. Classifications in species should not be based solely on morphological characteristics.

Answer: Phaeophyscia koreana is characterized by having a gray thallus, lacking soredia and isidia, lobules frequently present near the lobe and apothecia margin, apothecia with sparse rhizines near the base, white medulla, pale to white lower surface with a paraplectenchymatous cortex, absence of lichen substances. However, we deleted the two new species for the journal requirement.

Comment: The discussion is rather a recapitulation of the results. The authors should discuss in more detail the proposed taxonomic changes related to phylogenetic clades. In addition, this being a journal about microorganisms in general, the authors should discuss that lichens are a multispecies symbiosis (Aschenbrenner et al., 2016 10.3389/fmicb.2016.00180) and state the difficulties that this particularity entails for the classification of these organisms (Spribille 2018 10.1016/j.pbi.2018.02.007).

Answer: we have discussed more details in the discussion part.

Morberg [20] combined the species Physcia poeltii Frey into Phaeophyscia as Phaeophyscia poeltii (Frey) Nimis based on having a white lower surface and a prosoplectenchymatous lower cortex. According to our phylogenetic analysis, Phaeophyscia poeltii is in the Physciella clade, and by combining the cortex characters from Morberg’s identification [20], we propose treating the new combination as Physciella poeltii (Frey) D. Liu and J.S. Hur.

We also agree that lichens are multispecies symbiosis, but we think it is little related with our study. We briefly mentioned the background information on the lichen symbiosis in introduction section.

Reviewer 2 Report

This paper revises the lichen genus Phaeophyscia and several related taxa in South Korea. A total of 17 taxa are presented, including 2 new species, new combinations and new reports for Korea. Despite being foliose genera, these taxa are often critical: many species are often misunderstood and/or overlooked in many local and national flora, due to the difficult interpretation of several morphological characters. In my opinion, genera revisions like this manuscript are basic but fundamental contributions that help to link the taxonomic research to the potential application in the field of ecology.

The manuscript is well structured, well written and accurate. I particularly appreciate the iconography of the species, that is indeed of very high quality and useful to support the description of the main taxonomic characters. I add some detailed comments afterwards. Some changes are suggested to further improve the paper:

·      L10: why “approximately”? You could indicate the exact number.

·      L18 and following I suggest to give more emphasis to the new species and combinations, moving this result as “point a).

·      L36-37: this statement needs some references and explanations.

·      L49: for non-Korean, I suggest adding a couple of words for describing the bioclimate and/or habitat in Gangwon-do (montane, tropical?)

·      L51: I understand that this is nearly self-explaining, but I suggest adding a brief paragraph for stating formally the aim of the work here at the end of the introduction.

·      L61: Not sure to have got the TLC protocol. It seems a bit different from traditional protocols in lichenology (e.g. Culberson). Could you please explain it a little bit more?

·      In my opinion, a short description of main climatic and ecological features of Korea may help the reader to understand in what kind of habitats your species live.

·      Figure 1: Please, consider changing the color and/or the style of the bars for characters (e.g. atranorin…), to increase the clarity of the figure.

·      L120: I wonder if the spore type is the best character for differentiating Physconia from other genera in a genera key. Consider that some samples may not have apothecia… For identification purposes is preferable to rely on simple characters that are easier to detect. The same apply for L126-127 and L128-129.

·      L196: please, add a brief description of the macro-habitat (e.g. forest type, or agricultural area…). The same apply for other species.

Author Response

We very appreciated the editor and review’s comments to improve our manuscript.

We have revised our MS according to their comments and give our answer as following:

We marked corrections based on reviewers’ comment with yellow color.

We also marked corrections suggested from MDPI English-editing service with blue color.

Comment: L10: why “approximately”? You could indicate the exact number.

Answer: we change it to “with 17 species

Comment: L18 and following I suggest to give more emphasis to the new species and combinations, moving this result as “point a).

Answer: We have revised the abstract, and the new species were deleted for the journal requirement

Comment: L36-37: this statement needs some references and explanations.

Answer: we have added some reference.

This definition, however, was not fully accepted by all lichenologists [4,5,8].

Comment: L49: for non-Korean, I suggest adding a couple of words for describing the bioclimate and/or habitat in Gangwon-do (montane, tropical?)

Answer: we have added the bioclimate characters for this area.

South Korea is located in the southern half of the Korea Peninsula, and is dominated by mountains in the east and south and coastal plains and river tributary coastlines in the west. Gangwon-do(province) is located at northern part of South Korea along East sea (L.47-48)

Comment: L51: I understand that this is nearly self-explaining, but I suggest adding a brief paragraph for stating formally the aim of the work here at the end of the introduction.

Answer: We added it in L58-61.

In this study, we combined morphology, chemistry, and phylogenetic analysis with the aim of clarifying the relationship among the foliose genera in the Physciaceae family, and to investigate the constitution, distribution, and ecology of the atranorin-absent groups in Physciaceae more comprehensively.

Comment: L61: Not sure to have got the TLC protocol. It seems a bit different from traditional protocols in lichenology (e.g. Culberson). Could you please explain it a little bit more?

Answer: Because zeorine is a very important compound to identify the species of these groups, and norstictic acid is absent in these lichen genera. That’s the reason why we used zeorine as control, instead of norstictic acid.

Comment: In my opinion, a short description of main climatic and ecological features of Korea may help the reader to understand in what kind of habitats your species live.

Answer: we have added the geographical features. See line 45–46.

South Korea is located in the southern half of the Korea Peninsula, and is dominated by mountains in the east and south and coastal plains and river tributary coastlines in the west. (L.47-48)

Comment: Figure 1: Please, consider changing the color and/or the style of the bars for characters (e.g. atranorin…), to increase the clarity of the figure.

Answer: we have revised the figure 1 to black and white.

Comment: L120: I wonder if the spore type is the best character for differentiating Physconia from other genera in a genera key. Consider that some samples may not have apothecia… For identification purposes is preferable to rely on simple characters that are easier to detect. The same apply for L126-127 and L128-129.

Answer: it’s preferable to identify the specimens based on the simple characters, but in the foliose groups of Physciaceae, the spore, cortex types and the spot test is very useful for genera delimitation. In addition, there is little difference in other characteristics so far.

Comment: L196: please, add a brief description of the macro-habitat (e.g. forest type, or agricultural area…). The same apply for other species.

Answer: we would like to add the macro-habit, but most of the species were very common around Korea.

Reviewer 3 Report

I carefully went through the manuscript by Dong Liu and Jae-Seoun Hur dealing with the revision of the lichen genus Phaeophyscia and allied atranorin absent taxa (Physciaceae) in South Korea.

 This is a relevant and interesting topic, the number of specimens examined is appropiate and the work seems to me methodologically correct, the worldwide key is useful, however some revisions should be necessary to improve the clarity of the manuscript; in particular the manuscript in its current form is difficult to follow because of an inappropriate English style; a revision by a native speaker is strongly recommended.

General remarks:

-A general reference for the nomenclature for lichen species is lacking, authors should added a reference (e.g. Index fungorum)

- In the paragraph materials & methods authors should provide a legend for the data cited for each record: e.g State, province, city, valley, coordinates, substrate, date of collection, collector, ID number (for a non Korean reader it is difficult to distinguish geographical names).

A list of comments is subsequently reported to detail those points of the text which should/may be re-checked to improve the clarity of the contents.

1. Introduction

Line 44:(1980, 1981). Is not clear if this a citation of published record? Erbarium specimens? Authors can clarify this point?

Lines 46-48: “Hur (2005) listed 17 species of Phaeophyscia [43], but this was decreased to 16 species [41], and two species in Physciella in both publications.” Please clarify this point about Physciella

2. Materials and methods

Lines 59-60: “The ascospore dimensions range from 20–80 spores the size of a single apothecium per specimen.” I think that in this sentence some worlds are missing...

Lines 61: “special”: K, C, KC and P are not “special” solutions, these solutions are largely used in lichenology (cfr. Smith et al, 2009).

3. Results

Figure 1 (and other several occurrences) “prosplectenchymatous” is not the correct term: change in “prosoplectenchymatous”

3.3 Taxonomy

Line 112: ”Three genera (including 20 species), Hyperphyscia, Phaeophyscia, Physciella were confirmed” this sencence should be changed as follows: “the three genera (including 20 species) Hyperphyscia, Phaeophyscia and Physciella were confirmed;”

Line 115: ”Three species”: the excluded species are four

Line 116: “P. imbricate” is not a valid species, perhaps is P.imbricata?

Line  141: “concolor with a lower surface”  should be ““concolor with the lower surface”

Line  143: “algae cell” should be “algal cells”

Line 148-149: “(17.5)17.8–21.4–27.6(28.8)×(8.6)8.9–11.4–14.5(15.2) μm, Q = 1.6–2.6, Q(m)= 1.9, n = 44” can authors better clarify these measure of spores?

Line 155:”near the lobe” can authors clarify?

Line 178: “soradia” ...soredia? ...Soralia?

Line 182: “algae layer” should be “algal layer” (check throughout the manuscript)

Line 3.3.2. “New records” I suggest for this paraph the title “species new to South Korea”

Line 224: “Phaeophyscia hunana“ should be “Phaeophyscia hunnana”

Line 225: “obtained” maybe “found”?

Lines 291-292: “and first time reported from South Korea.” should be “and ere first time reported from South Korea.”

Line 814: “concentrate together” this expression is not clear to me…

Line 825:”H. confuse”  should be “  “H. confusa”

Line 828: “soralia laminar” should be “laminar soralia”

Line 830:”information is difficult to obtain because of the lower support.” can authors clarify?

Line 837:”and lobe smaller” should be “and its lobes smaller”

Line 838:”actually contain two species” perhaps “are assigned to different species”?

854-955 “Worldwide key to species of Phaeophyscia and Physciella” the key are usefull (i’ve tried identification of some European species) but  the English is very rough.

Line 863:”lobulates” should be “lobules” (check throughout the key)

Line 872:”rhizine”  should be ”rhizines”

Lines 874-875: “moss over rock” should be “saxicolous mosses”

Line 893:”Upper surface or lobe end more or less pruina” should be “Upper surface or lobes end more or less pruinose”

Line 903: “P. squarossa”--->”P. Squarrosa”

Line 916: “soralia lip-shape” should be soralia lip-shaped”

Line 941: “Upper surface without maculate” should be “Upper surface without maculae”

Supplementary materials:

-Table S1. Is missing...(??)

- the List S1. Is not clear: maybe authors can re-organize the information as table?

p { margin-bottom: 6.25px; line-height: 120%; }

Author Response

We very appreciated the editor and review’s comments to improve our manuscript.

 We have revised our MS according to their comments and give our answer as following:

 We marked corrections based on reviewers’ comment with yellow color.

We also marked corrections suggested from MDPI English-editing service with blue color.

Comment: a revision by a native speaker is strongly recommended.

Answer: Professional English-editing service was conducted by MPDI (The edited MS can be downloaded from author submission page in MPDI system)

General remarks:

Comment: A general reference for the nomenclature for lichen species is lacking, authors should added a reference (e.g. Index fungorum)

Answer: we have changed the version of the nomenclature for lichen species. And we simplified the name without the reference, which will shorten the manuscript to a more manageable length.

Comment: In the paragraph materials & methods authors should provide a legend for the data cited for each record: e.g State, province, city, valley, coordinates, substrate, date of collection, collector, ID number (for a non Korean reader it is difficult to distinguish geographical names).

Answer: we have changed the data into English address style. The changes were marked with yellow color. For example, Gyeongsangnam Province, Hamyang County, Seosang Town.

1. Introduction

Comment: Line 44:(1980, 1981). Is not clear if this a citation of published record? Erbarium specimens? Authors can clarify this point?

Answer: we have added the literatures and altered the order of reference.

Since the first species of Phaeophyscia endococcina (= P. endococcinodes) was recorded in Korea [30,31],

Comment: Lines 46-48: “Hur (2005) listed 17 species of Phaeophyscia [46], but this was decreased to 16 species [41], and two species in Physciella in both publications.” Please clarify this point about Physciella

Answer: we have revised it as following. L. 52-54.

Hur listed 17 species of Phaeophyscia [46]; later, this number decreased to 16 species in Moon[KO1]  [44], and both publications recorded two species of Physciella.

2. Materials and methods

Comment: Lines 59-60: “The ascospore dimensions range from 20–80 spores the size of a single apothecium per specimen.” I think that in this sentence some worlds are missing...

Answer: we changed it to “from a single apothecium per specimen.” L. 72

The ascospore dimensions ranged from 20–80 spores from a single apothecium per specimen.

Comment: Lines 61: “special”: K, C, KC and P are not “special” solutions, these solutions are largely used in lichenology (cfr. Smith et al, 2009).

Answer: we changed it to “ a solution” L. 73.

3. Results

Comment: Figure 1 (and other several occurrences) “prosplectenchymatous” is not the correct term: change in “prosoplectenchymatous”

Answer: we checked it throughout the manuscript and changed to “prosoplectenchymatous”

3.3 Taxonomy

Comment: Line 112: ”Three genera (including 20 species), Hyperphyscia, Phaeophyscia, Physciella were confirmed” this sencence should be changed as follows: “the three genera (including 20 species) Hyperphyscia, Phaeophyscia and Physciella were confirmed;”

Answer: we revised it according to the comment, but we deleted the two new species, so that we included 18 species, see L. 133.

Three genera (including 18 species), Hyperphyscia, Phaeophyscia, Physciella, were confirmed;

Comment:Line 115: ”Three species”: the excluded species are four

Answer: we changed it to “four species” see Line 136

Comment: Line 116: “P. imbricate” is not a valid species, perhaps is P.imbricata?

Answer: we changed it to Phaeophyscia imbricata[KO2] , see line 137

Comment: Line 141: “concolor with a lower surface” should be “concolor with the lower surface”

Answer: we deleted tow new species because of journal requirement.

Comment: Line 143: “algae cell” should be “algal cells”

Answer: we deleted tow new species.

Comment: Line 148-149: “(17.5)17.8–21.4–27.6(28.8)×(8.6)8.9–11.4–14.5(15.2) μm, Q = 1.6–2.6, Q(m)= 1.9, n = 44” can authors better clarify these measure of spores?

Answer: we deleted tow new species.

(Minimum)range-mean value-(maximum). The number (N) of spores were measured, and mean values (in italics), quotient of length and width (Q), and average quotient Q(m) were calculated.

Comment: Line 155:”near the lobe” can authors clarify?

Answer: we deleted tow new species.

that means the lobules present margin or submargin.

Comment: Line 178: “soradia” ...soredia? ...Soralia?

Answer: we deleted tow new species.

it’s soredia, anyway, we deleted this species in this paper.

Comment: Line 182: “algae layer” should be “algal layer” (check throughout the manuscript)

Answer: we deleted tow new species. we checked throughout the manuscript and changed it to “algal layer”

Comment: Line 3.3.2. “New records” I suggest for this paraph the title “species new to South Korea”

Answer: we change it to “species new to South Korea”. L. 155

Comment: Line 224: “Phaeophyscia hunana“ should be “Phaeophyscia hunnana”

Answer: the name should be Phaeophyscia hunana

Comment: Line 225: “obtained” maybe “found”?

Answer: we revised it. L. 168. and here reported for the first time in South Korea.

Comment: Line 814: “concentrate together” this expression is not clear to me…

Answer: we changed it to “gathered together”, (L.747) that means the foliose genera forming a broad clade and separated from the crustose groups.

Comment:  Line 825:”H. confuse”  should be “  “H. confusa”

Answer: we changed it to Hyperphyscia confusa.” L. 761.

Comment: Line 828: “soralia laminar” should be “laminar soralia”

Answer: we changed it to “laminar soralia”. L.764.

Comment: Line 830:”information is difficult to obtain because of the lower support.” can authors clarify?

Answer: in the clade of Phaeophyscia, the inner branch support value is low. The medulla difference is clear from morphology, but medulla species and orange medulla species mixed in the phylogenetic tree with lower suport. Therefore we changed it to “however, more information was difficult to obtain due to the lesser support.”

Comment: Line 837:”and lobe smaller” should be “and its lobes smaller”

Answer: we changed it toand had smaller lobes”, see line 774.

Comment: Line 838:”actually contain two species” perhaps “are assigned to different species”?

Answer: yes, we changed it to “were assigned to different species”, see line 775.

Comment: 854-955 “Worldwide key to species of Phaeophyscia and Physciella” the key are usefull (i’ve tried identification of some European species) but the English is very rough.

Answer: we intensively revised the key, and see the changes marked with yellow color.

Comment: Line 863:”lobulates” should be “lobules” (check throughout the key)

Answer: we checked it to “lobules” throughout the key.

Comment: Line 872:”rhizine”  should be ”rhizines”

Answer: we checked it to rhizinesthroughout the key.

Comment: Lines 874-875: “moss over rock” should be “saxicolous mosses

Answer: we checked it to change to “saxicolous mosses”.

Comment: Line 893:”Upper surface or lobe end more or less pruina” should be “Upper surface or lobes end more or less pruinose

Answer: we changed it to “Upper surface or lobes end more or less pruinose”.

Comment: Line 903: “P. squarossa”--->”P. Squarrosa”

Answer: we changed it toP. squarrosa.”

Comment: Line 916: “soralia lip-shape” should be soralia lip-shaped”

Answer: we changed it to “soralia lip-shaped”.

Comment: Line 941: “Upper surface without maculate” should be “Upper surface without maculae

Answer: we changed it toUpper surface without maculae”.

Supplementary materials:

Comment: Table S1. Is missing...(??)

Answer: we added the table S1

Comment: the List S1. Is not clear: maybe authors can re-organize the information as table?

Answer: S1 is the specimen information (location, substrate and collection number, etc.), and we have revised the List S1, the places where changed have been marked as yellow background.

Comment: Margin bottom: 6.25px; line-height: 120%

Answer: we have change the format. 

Round 2

Reviewer 1 Report

The manuscript improved considerably since the previous review, I thank the authors for having taken all the suggestions into account. Since descriptions of new species were eliminated in the new version, now I have no further objections to the manuscript. Just note that the scientific names that are used as subtitles of the subsections should be in italics.